# Auditory cortex shapes sound responses in the inferior colliculus

**Jennifer M Blackwell[1,2], Alexandria MH Lesicko[1], Winnie Rao[1], Mariella De Biasi[3,4,5], Maria N Geffen[1,5,6]\***

[1]Department of Otorhinolaryngology, University of Pennsylvania, Philadelphia, United States; [2]Department of Neurobiology and Behavior, Stony Brook University, Stony Brook, United States; [3]Department of Psychiatry, University of Pennsylvania, Philadelphia, United States; [4]Department of Systems Pharmacology and Experimental Therapeutics, University of Pennsylvania, Philadelphia, United States; [5]Department of Neuroscience, University of Pennsylvania, Philadelphia, United States; [6]Department of Neurology, University of Pennsylvania, Philadelphia, United States

**Abstract** The extensive feedback from the auditory cortex (AC) to the inferior colliculus (IC) supports critical aspects of auditory behavior but has not been extensively characterized. Previous studies demonstrated that activity in IC is altered by focal electrical stimulation and pharmacological inactivation of AC, but these methods lack the ability to selectively manipulate projection neurons. We measured the effects of selective optogenetic modulation of cortico-collicular feedback projections on IC sound responses in mice. Activation of feedback increased spontaneous activity and decreased stimulus selectivity in IC, whereas suppression had no effect. To further understand how microcircuits in AC may control collicular activity, we optogenetically modulated the activity of different cortical neuronal subtypes, specifically parvalbumin-positive (PV) and somatostatin-positive (SST) inhibitory interneurons. We found that modulating the activity of either type of interneuron did not affect IC sound-evoked activity. Combined, our results identify that activation of excitatory projections, but not inhibition-driven changes in cortical activity, affects collicular sound responses.

**\*For correspondence:**
mgeffen@pennmedicine.upenn.edu

**Competing interests:** The authors declare that no competing interests exist.

## Introduction

Information processing is typically studied as a set of computations ascending along a hierarchy of sensory areas, often overlooking the role of descending feedback between these nuclei. In the auditory system, the auditory cortex (AC) sends extensive feedback to nuclei earlier in the auditory pathway, including the auditory thalamus (*Alitto and Usrey, 2003*; *Rouiller and Durif, 2004*; *Winer et al., 2001*) and the inferior colliculus (IC) (*Bajo and Moore, 2005*; *Bajo et al., 2007*; *Coomes et al., 2005*; *Doucet et al., 2003*; *Saldaña et al., 1996*; *Williamson and Polley, 2019*). Previous studies have demonstrated the importance of cortical feedback to IC in auditory behaviors (*Bajo et al., 2010*; *Xiong et al., 2015*), however, the mechanisms by which information processing is shaped via the *descending feedback* pathway remain poorly characterized. In this study, we selectively modulated activity in AC, targeting either excitatory projections or inhibitory interneurons, to quantify how AC shapes spontaneous activity and sound-evoked responses in IC.

Previous studies demonstrated that neuronal responses to sounds in IC are altered by focal electrical stimulation and inactivation of AC. Cortical stimulation shifted tuning properties of IC neurons toward those of the stimulated neurons in frequency (*Jen et al., 1998*; *Jen and Zhou, 2003*; *Ma and Suga, 2001a*; *Yan et al., 2005*; *Yan and Suga, 1998*; *Zhou and Jen, 2007*), amplitude (*Jen and Zhou, 2003*; *Yan et al., 2005*; *Zhou and Jen, 2007*), azimuth (*Zhou and Jen, 2007*;

**eLife digest** How do we hear the world around us? Hearing begins when hair cells in the inner ear translate incoming sound waves into electrical signals. These signals travel via the auditory nerve and the brainstem to the midbrain, where an area called the inferior colliculus processes them. The inferior colliculus then passes the signals on to another area deep within the brain, the thalamus, which processes the signals further before it too passes them on to an area of the brain's outer layer called the auditory cortex.

At each stage of the auditory pathway, the signals undergo more complex processing than at the previous stage. Researchers have tended to think of this pathway as a one-way route from the ear to the brain. But in reality, feedback occurs at various points along the pathway, enabling areas that do higher processing to shape the responses of areas earlier in the pathway. This feedback is particularly prevalent in the auditory system, where one such strong feedback route is from the auditory cortex to the inferior colliculus. This reverse connection helps animals learn new behavioral responses to sounds, for example, to run away from a loud noise.

By manipulating the activity of this pathway in mice using a technique called optogenetics, Blackwell et al. provide further clues to how the auditory pathway works. Optogenetics involves introducing light-sensitive ion channels into neurons, and then using light to activate or inhibit those neurons on demand. Blackwell et al. show that activating the feedback pathway from the auditory cortex to the inferior colliculus in awake mice changes how the inferior colliculus responds to sounds. By contrast, inhibiting the pathway has no effect on inferior colliculus responses. This suggests that the feedback pathway is not active all the time, but instead influences inferior colliculus activity only during specific behavior, for example, perhaps when we are listening for a specific sound like the ringing of a phone.

Understanding how the brain processes sound is important for understanding how we communicate and why we appreciate music. It could also help in treating hearing loss. Stimulating the inferior colliculus using a device implanted in the brainstem can improve hearing in people with certain types of deafness. Strengthening or weakening the feedback pathway from the auditory cortex to the inferior colliculus could make these implants more effective. In the future, it may even be possible that stimulating the pathway directly could restore hearing without any implant being required.

---

*Zhou and Jen, 2005*), and duration (*Ma and Suga, 2001b*). Stimulation of AC had mixed effects on sound-evoked responses in IC, increasing and decreasing responses in different subpopulations of neurons (*Jen et al., 1998*; *Zhou and Jen, 2005*). Consistent with this effect, different patterns of direct cortico-collicular activation enhanced or suppressed white noise-induced responses in IC (*Vila et al., 2019*). AC inactivation studies, on the other hand, found less consistent effects on IC responses. Whereas one study found that inactivation of AC caused a shift in best frequency in IC neurons (*Zhang et al., 1997*), several other studies showed that inactivation of AC had no effect on frequency selectivity in IC (*Jen et al., 1998*), but rather modulated sound-evoked and spontaneous activity (*Gao and Suga, 1998*; *Popelár et al., 2003*; *Popelář et al., 2016*). Cortico-collicular feedback is critical to auditory learning, specifically learning to adapt to a unilateral earplug during sound localization (*Bajo et al., 2010*). Pairing electrical leg stimulation with a tone induced a shift in best frequency of IC neurons, while presentation of a tone alone was insufficient (*Gao and Suga, 1998*; *Gao and Suga, 2000*). Furthermore, cortico-collicular feedback was necessary to induce running in response to a loud noise (*Xiong et al., 2015*).

In AC, modulation of sound responses is not a monophasic process, but instead is shaped by interactions between excitatory and inhibitory cortical neurons. Modulating activity of different inhibitory interneuron subtypes in AC narrowed frequency tuning and attenuated tone-evoked responses of excitatory neurons, while suppression had the opposite effect (*Aizenberg et al., 2015*; *Hamilton et al., 2013*; *Phillips and Hasenstaub, 2016*; *Seybold et al., 2015*). Electrical stimulation of AC, cooling or pharmacological inactivation affected the amplitude of sound-evoked responses and shifted the best frequency of neurons in IC, but it remains unknown how specific these effects are to direct feedback, and whether the effects of intra-cortical inhibition propagate to the IC.

The goal of the present study is to examine the role of cortico-collicular projections in shaping sound responses in IC. IC receives glutamatergic (*Feliciano and Potashner, 1995*) inputs from neurons originating predominantly in layer 5 of AC (*Bajo and Moore, 2005*; *Bajo et al., 2007*; *Coomes et al., 2005*; *Doucet et al., 2003*; *Saldaña et al., 1996*; *Winer et al., 1998*). We used viral transfection methods to selectively drive excitatory or inhibitory opsin expression in AC-IC projections. We then recorded neuronal activity in IC and tested how activation or suppression of AC-IC projections affected spontaneous activity and sound-evoked responses in IC. To better understand whether and how intra-cortical network interactions propagated to IC, we manipulated the activity of the two most common inhibitory neuronal subtypes in AC, Parvalbumin-(PV) and Somatostatin-(SST) positive interneurons (*Rudy et al., 2011*).

## Results

### Activating direct cortico-collicular feedback modulates activity in the inferior colliculus

Our first goal was to characterize the effects of activating the direct cortico-collicular projections on tone-evoked responses in IC. A combination of cortical anterograde and collicular retrograde viral transfections was used in order to achieve specific viral transfection of cortico-collicular neurons. We delivered either an excitatory opsin, ChannelRhodopsin2 (ChR2) or an inhibitory opsin, ArchaerhodopsinT (ArchT), bilaterally, specifically to the neurons in the auditory cortex which project to the inferior colliculus (*Figure 1A*). To achieve such specificity, we injected a retrograde virus that encoded Cre recombinase (Retro2 AAV.Cre) in IC (*Figure 1B*). This retrograde viral construct ensured that neurons projecting to IC expressed Cre recombinase a few weeks later. At the same time, we injected a virus that encoded ChR2 or ArchT in reversed fashion under the FLEX cassette in AC (AAV.Flex.ChR2, AAV.Flex.ArchT) (*Figure 1C*). This strategy ensured that only neurons expressing Cre recombinase in the auditory cortex would express ChR2 or ArchT in AC. Therefore, opsin was expressed exclusively in AC-IC projecting neurons, which terminate across all regions of IC, including central nucleus (CNIC) and dorsal cortex of IC (DCIC), from which we recorded (*Figure 1D*). Shining light over AC of these mice would therefore directly activate or suppress only these cortico-collicular feedback projections to IC.

First, we measured and quantified neuronal spiking in IC in response to stimulation or suppression of cortico-collicular projections using ChR2 or ArchT. To activate this projection, we shone blue laser over AC, recorded neuronal activity in IC, and quantified the effects of manipulating feedback in the absence of sound (*Figure 2A,C,D*). We measured the spiking activity in IC as we varied the duration of laser manipulation (1 ms, 5 ms, 25 ms, 250 ms). As exected, activation of cortico-collicular neurons resulted, on average, in an increase in firing rate of IC neurons. This effect persisted at the 5 ms, 25 ms, and 250 ms laser duration (*Figure 2C*; *1 ms*: p=5.3e-4, ON = 5.6 ± 0.6 Hz, OFF = 4.9 ± 0.5 Hz; *5 ms*: p=0.0027, ON = 6.9 ± 0.7 Hz, OFF = 4.8 ± 0.5 Hz; *250 ms*: p=0.005, ON = 5.6 ± 0.6 Hz, OFF = 4.8 ± 0.5 Hz). Whereas the direction of the effect was consistent with our prior expectations, the magnitude of the effect was unexpectedly small. This suggests that AC targets a small subpopulation of IC neurons and that the effect of activation does not spread far within IC.

We tested the effect of suppressing AC-IC neurons on firing responses in IC and, surprisingly, we detected little difference in firing in IC neurons (*Figure 2D*; *25 ms*: p=0.039, ON = 9.5 ± 0.7 Hz, OFF = 9.5 ± 0.7 Hz; *250 ms*: p=0.02, ON = 9.5 ± 0.8 Hz, OFF = 9.5 ± 0.8 Hz). This result suggests that AC-IC inputs at rest do not contribute to IC responses, but rather modulate collicular activity only when activated.

Next, our goal was to characterize modulation of sound-evoked responses in IC by cortical feedback. We first tested the effects of feedback modulation on acoustic click responses. We chose clicks as the initial stimulus because they drive fast responses in both AC and IC. Laser stimulation began 250 ms prior to click train onset to allow for the response to the laser to come to a steady state, and continued activation throughout the clicks (*Figure 2B*). Activating the feedback using ChR2 had a weak suppressive effect on the firing rate of IC neurons in response to clicks. Consistent with previous experiments, we observed an overall increase in spontaneous activity (*Figure 2E*, bottom; p=0.0031, spont ON = 5 ± 0.87 Hz, spont OFF = 4.3 ± 0.92 Hz). Activating feedback caused a small

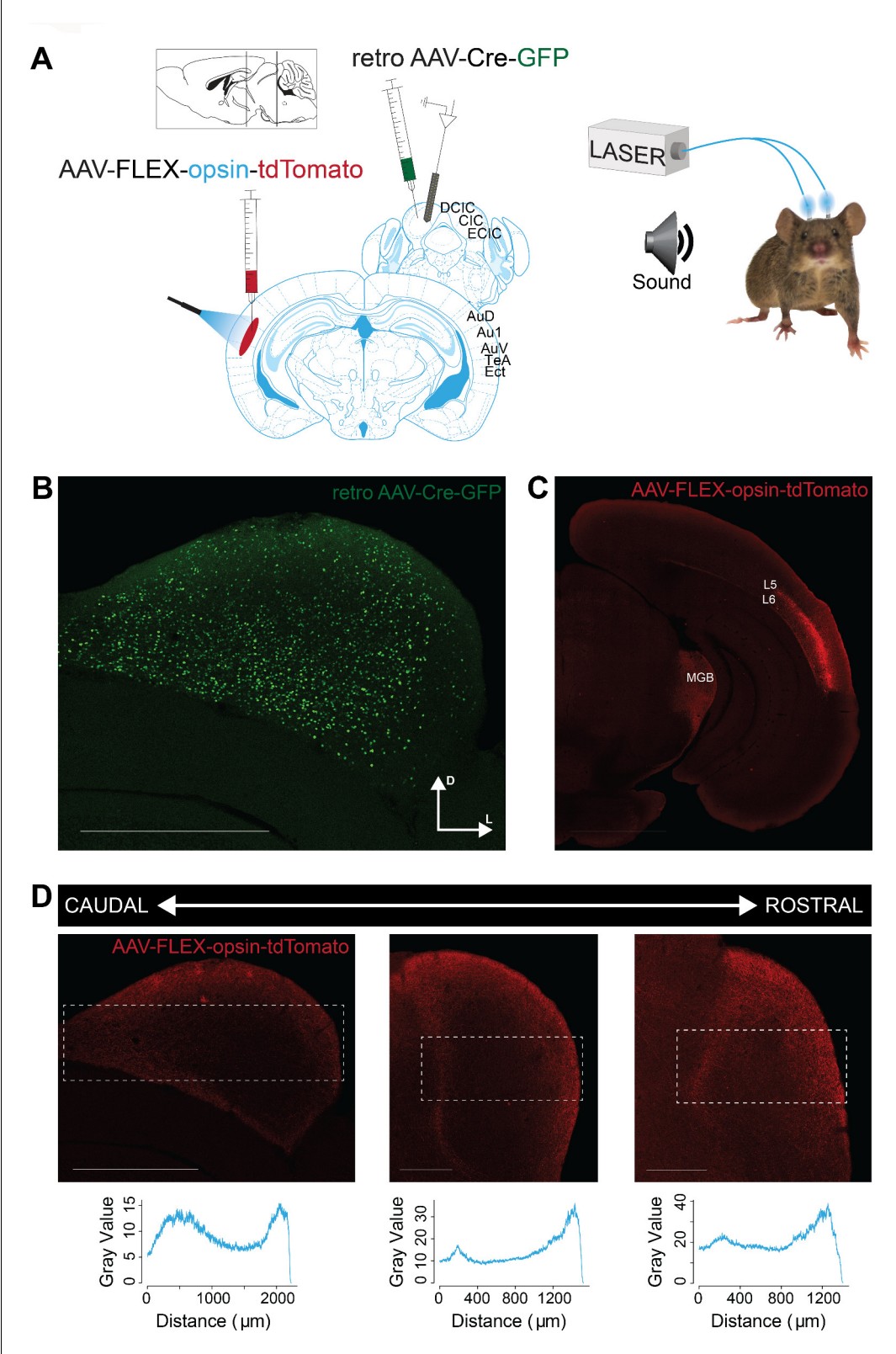

**Figure 1.** Opsin expression in corticocollicular feedback. (**A**) Experimental design: To yield selective labeling of auditory cortico-collicular projections, a retroAAV-Cre-GFP construct is injected into the IC and AAV-FLEX-opsin-tdTomato construct is injected into the AC bilaterally. Recordings are made in multiple bilateral locations of the IC (left) of awake mice. Cannulas are implanted bilaterally in the auditory cortex, and a green/blue laser is used to silence/activate the cortico-collicular pathway (right). (**B**) Expression of the retro AAV-Cre-GFP construct in the IC. Scale bar: 1000 µm. (**C**) Selective

*Figure 1 continued on next page*

*Figure 1 continued*

expression of the AAV-FLEX-opsin-tdTomato construct in cell bodies in layer 5 (L5) and 6 (L6) of the AC that project to the IC. Fiber and terminal labeling is also present in the medial geniculate body, a known target of auditory cortico-collicular cells. Scale bar: 2000 μm. (D) Expression of the AAV-FLEX-opsin-tdTomato construct in fibers and terminals at multiple rostro-cadual levels of the IC. Expression, as measured by fluorescence intensity (bottom), is strongest in medial portions of the central nucleus of the IC and lateral portions of the external nucleus of the IC. Scale bar: left, 1000 μm; middle and right, 500 μm.

decrease in click-evoked response (*Figure 2E*, left; p=0.001, click ON = 10.4 ± 1.2 Hz, click OFF = 10.7 ± 1.1 Hz).

By contrast, suppressing cortico-collicular feedback using ArchT had no effect on IC click responses (*Figure 2F*). Because attenuating the input does not affect the activity in IC during clicks, this result further suggests that the strength of the baseline signal from AC to IC, even in the presence of cortical activity, is not sufficient to modulate IC responses.

We also tested whether the effects of cortico-collicular modulation differed across tone frequency range. We presented a stimulus that consisted of tones of 50 frequencies ranging from 3 to 70 kHz. To modulate the feedback from AC, we tested three different laser onsets (−100 ms, −20 ms, +8 ms) relative to tone onset to isolate the effect of timing on affecting IC responses. These delays were chosen for the following reasons: −100 ms delay would allow for the laser effect on cortical activity to come to a steady state, allowing to quantify the effect throughout the tone pip; +8 ms is set up to mimic the time scale of cortical response to a tone, effectively amplifying the onset of the cortical response; −20 ms delay is an intermediate value.

We found that activating cortico-collicular feedback using ChR2 increased spontaneous activity of IC neurons (*Figure 3A*, left; −100 ms: p=0.022, ON = 4.3 ± 0.47 Hz; OFF = 3.6 ± 0.38; −20 ms: p=8.6e-6, ON = 5.6 ± 0.63, OFF = 3.7 ± 0.47). However, overall the feedback decreased tone-evoked response magnitude in IC, which we defined as the difference between spontaneous and tone-evoked response, at all laser onsets (*Figure 3A*, right; −100 ms: p=3.2e-5, ON = 14.2 ± 0.98 Hz; OFF = 15.5 ± 1.04 Hz; −20 ms: p=1.4e-6, ON = 11.8 ± 1.08 Hz, OFF = 13.9 ± 1 Hz; +8 ms: p=0.0034, ON = 13.03 ± 1.03, OFF = 13.7 ± 1.03). This suggests that the broad activation of feedback upregulates the baseline activity of IC neurons, but decreases tone-evoked response (Figure 6C, *−100* ms: p=0.018, ON = 19.01 ± 1.2 Hz, OFF = 18.5 ± 1.2 Hz; *+8* ms: p=0.007, ON = 17.3 ± 1.2 Hz; OFF = 16.9 ± 1.2 Hz). By contrast, suppressing the feedback using ArchT had no effect on either spontaneous activity or tone-evoked response magnitude (*Figure 3B*), suggesting that at baseline AC does not provide strong modulation of IC activity, as removing it does not affect sound-evoked effects in IC.

We then examined the effect of feedback on frequency tuning of IC units. Activation of feedback using ChR2 decreased frequency selectivity in the subsets of IC units that exhibited a decrease in tone-evoked response magnitude or increase in spontaneous activity (*Figure 4A*), but not in units that exhibited an increase in tone-evoked response magnitude or decrease in spontaneous activity (*Figure 4B*, mag decrease: −20 ms, p=0.00031, sparse ON = 0.49 ± 0.027, sparse OFF = 0.55 ± 0.025; +8 ms, p=0.00029, sparse ON = 0.46 ± 0.027, sparse OFF = 0.55 ± 0.024; spont increase: −100 ms, p=0.011, sparse ON = 0.49 ± 0.032, sparse OFF = 0.56 ± 0.031; −20 ms, p=0.00018, sparse ON = 0.46 ± 0.034, sparse OFF = 0.56 ± 0.029; +8 ms, p=2.2e-6, sparse ON = 0.41 ± 0.032, sparse OFF = 0.55 ± 0.029).

Interestingly, these changes were observed almost exclusively in cells located in DCIC rather than CNIC (*Figure 5*). Specifically, activation of feedback increased spontaneous activity (*Figure 5A*, left; *−100 ms*: p=0.0204, ON = 3.5 ± 0.5 Hz, OFF = 2.5 ± 0.3 Hz; *−20 ms*: p=2.8e-7, ON = 5.2 ± 0.7 Hz, OFF = 2.3 ± 0.3 Hz) and decreased tone-evoked response magnitude (*Figure 5A*, right, *−100 ms*: p=3.3e-6, ON = 14.8 ± 1.4 Hz, OFF = 16.5 ± 1.5 Hz; *−20 ms*: p=6.3e-7, ON = 11.8 ± 1.6 Hz, OFF = 14.7 ± 1.4 Hz) in cells located in DCIC, but not in CNIC (*Figure 5B*). Similarly, activation of feedback decreased frequency selectivity (*Figure 5C*; mag decrease: −20 ms, p=6.3e-4, sparse ON = 0.46 ± 0.03, sparse OFF = 0.53 ± 0.03; +8 ms, p=4.7e-5, sparse ON = 0.41 ± 0.03, sparse OFF = 0.53 ± 0.03; spont increase: −100 ms, p=0.0044, sparse ON = 0.43 ± 0.04, sparse OFF = 0.52 ± 0.04; −20 ms, p=0.00013, sparse ON = 0.39 ± 0.04, sparse OFF = 0.53 ± 0.04; +8 ms, p=5.2e-6, sparse ON = 0.33 ± 0.03, OFF = 0.53 ± 0.04) in cells located in DCIC, but not in CNIC (*Figure 5D*). This is consistent with the enhanced density of AC projections in DCIC as compared to

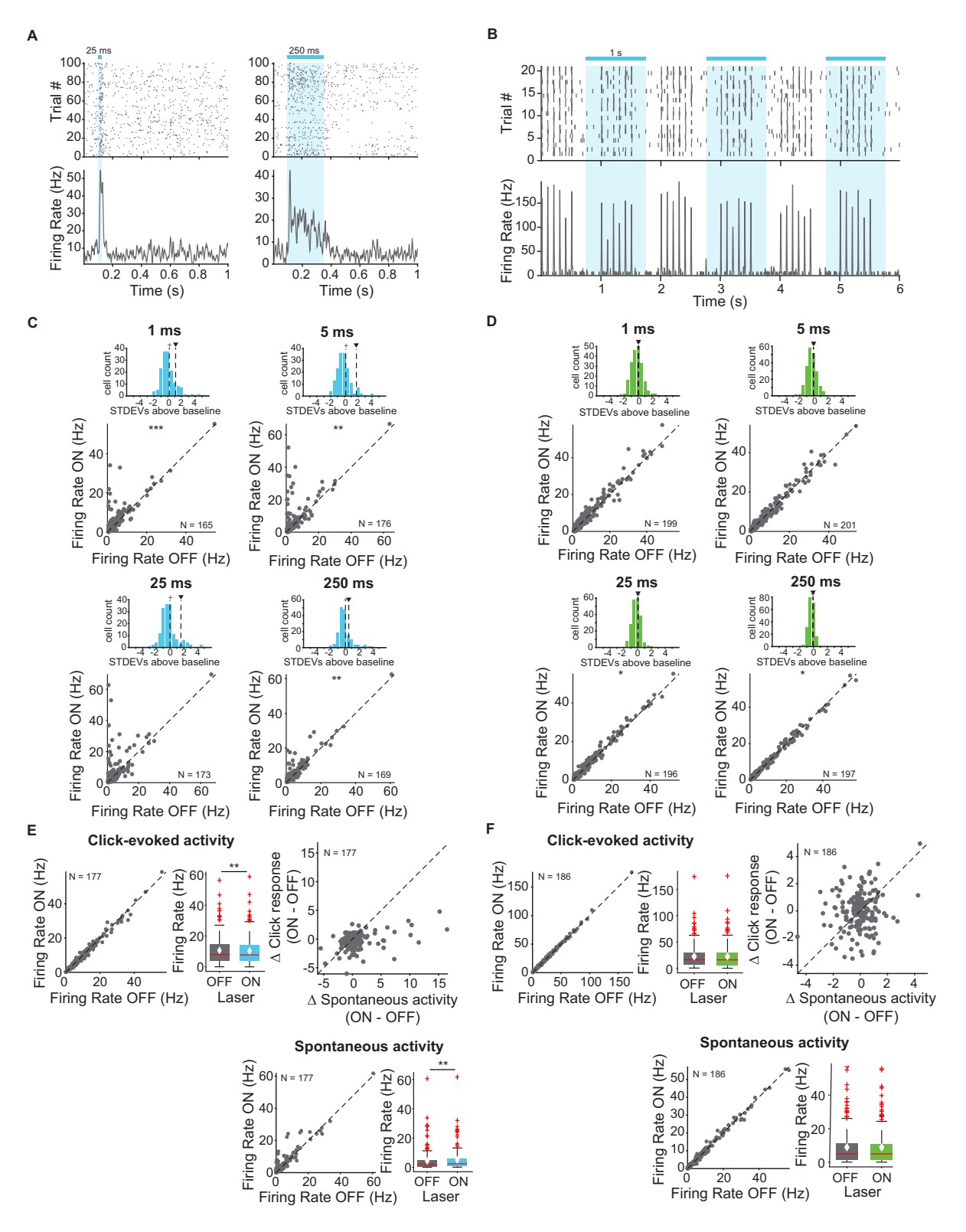

**Figure 2.** Effects of activating and inactivating cortico-collicular feedback activity on firing rate of neurons in IC in the absence of sounds and in response to sound clicks. (**A**) Example IC unit response to two laser pulse durations, activation (ChR2) (blue), 25 ms (left) and 250 ms (right). Top: raster plot of spike times. Bottom: average firing rate. Blue bar indicates the timing of the laser. (**B**) Example unit response to clicks with (blue) and without laser (activation, ChR2). Axes same as A. We presented 6 clicks at 10 Hz every second, starting at 0 s. (**C**) Change in IC unit activity in response

*Figure 2 continued on next page*

*Figure 2 continued*

to different laser durations with cortico-collicular acitvation using ChR2 (1 ms, 5 ms, 25 ms, 250 ms). *Top panels*: Histogram of changes in unit activity between laser ON and laser OFF conditions, + indicates median, ∨ indicates mean. These distributions are not normally distributed (Kolmogorov-Smirnov test; *1 ms*: p=3.07e-15, mean = 1.1 stdevs, median = 0.21 stdevs; *5 ms*: p=4.4e-20, mean = 1.7 stdevs, median = 0.19 stdevs; *25 ms*: p=5.5e-17, mean = 1.6 stdevs, median = −0.021 stdevs; *250 ms*: p=3.4e-12, mean = 0.58 stdevs, median = 0.101 stdevs). *Bottom panels*: Firing rate of IC units in laser ON versus laser OFF conditions (Statistics for the difference across the neuronal population: *1 ms*: p=5.3e-4, ON = 5.6 ± 0.6 Hz, OFF = 4.9 ± 0.5 Hz; *5 ms*: p=0.0027, ON = 6.9 ± 0.7 Hz, OFF = 4.8 ± 0.5 Hz; *250 ms*: p=0.005, ON = 5.6 ± 0.6 Hz, OFF = 4.8 ± 0.5 Hz)). (D) Change in IC unit activity in response to different laser durations with cortico-collicular inactivation using ChR2 (1 ms, 5 ms, 25 ms, 250 ms). *Top panels:* Histogram of changes in unit activity between laser ON and laser OFF conditions, + indicates median, ∨ indicates mean. These distributions are normally distributed (Kolmogorov-Smirnov test). Bottom panels: Firing rate of units in laser ON versus laser OFF conditions (Statistics for the difference across the neuronal population: *25 ms*: p=0.039, ON = 9.5 ± 0.7 Hz, OFF = 9.5 ± 0.7 Hz; *250 ms*: p=0.02, ON = 9.5 ± 0.8 Hz, OFF = 9.5 ± 0.8 Hz). (E) IC population responses to clicks exhibit a decrease in click response with cortico-cllicular activation using ChR2 (left; p=0.001, click ON = 10.4 ± 1.2 Hz, click OFF = 10.7 ± 1.1 Hz) and an increase in spontaneous activity (bottom; p=0.0031, spont ON = 5 ± 0.87 Hz, spont OFF = 4.3 ± 0.92 Hz). Left panel: Click-evoked firing rate of IC units on laser ON versus laser OFF conditions. Center panel: box plot of change in click-evoked firing rate On box plots, red line indicates median, white ◆ indicates mean. Right panel: Change in click response versus change in spontaneous activity due to laser activation. Bottom panel: Spontaneous activity on laser ON versus laser OFF trials. (F) IC population responses to clicks exhibit no change in activity with cortico-collicular suppression using ArchT. On box plots, red line indicates median, white ◆ indicates mean. Axes as in E.

CNIC (*Figure 1D*). To verify that these results were not due to a select few high firing rates of units in DCIC, we repeated the analysis comparing only units with spontaneous activity in the laser OFF condition less than 40 Hz. With these matched firing rates we still observed the same statistically

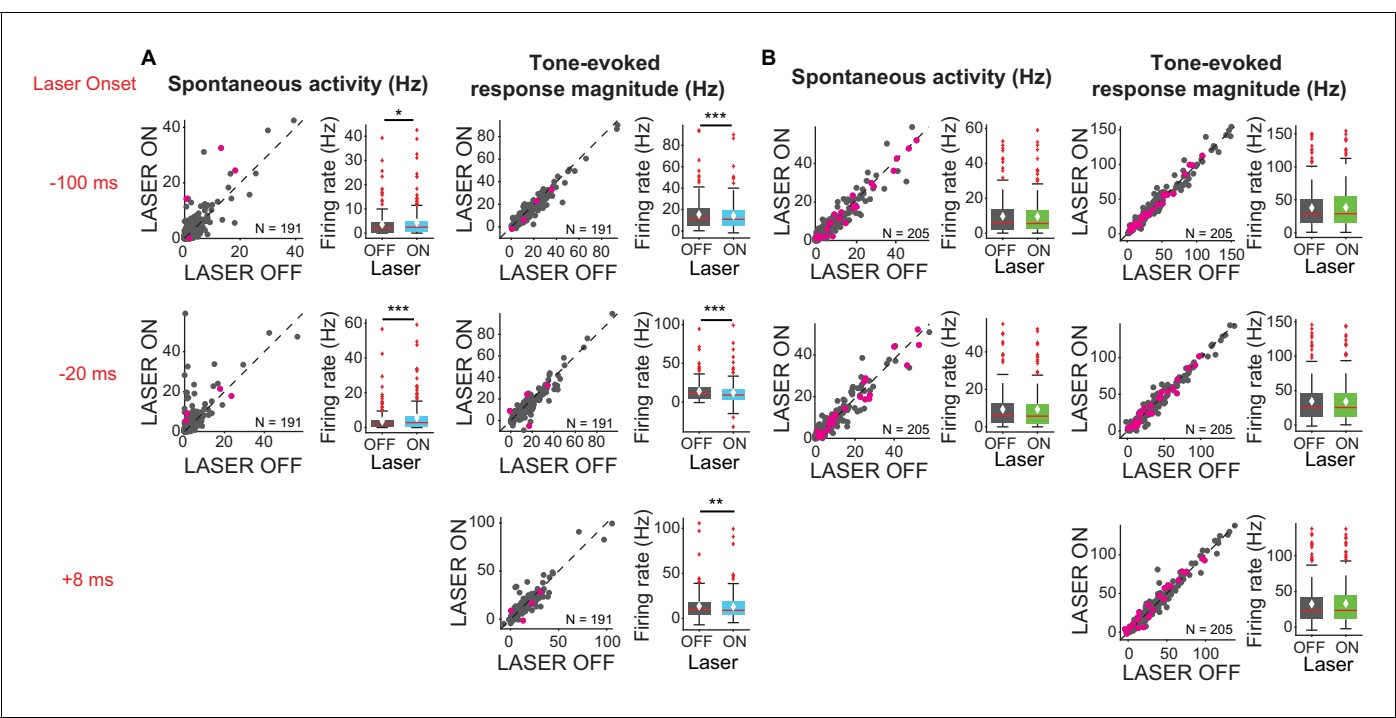

**Figure 3.** Effects of modulating feedback activity on spontaneous and tone-evoked response magnitude in IC. Left panels: neuronal activity (spontaneous, left or tone-evoked, right) on laser ON versus laser OFF trials. Right panels: Box plots of neuronal activity (spontaneous, left or tone-evoked, right) for laser ON and laser OFF trials. Magenta circles indicate single units. On box plots, red line indicates median, white ◆ indicates mean. (A) Activating feedback using ChR2 increased IC spontaneous activity (left; −100 ms: p=0.022, ON = 4.3 ± 0.47 Hz; OFF = 3.6 ± 0.38; −20 ms: p=8.6e-6, ON = 5.6 ± 0.63, OFF = 3.7 ± 0.47) and decreased tone-evoked response magnitude (right; −100 ms: p=3.2e-5, ON = 14.2 ± 0.98 Hz; OFF = 15.5 ± 1.04 Hz; −20 ms: p=1.4e-6, ON = 11.8 ± 1.08 Hz, OFF = 13.9 ± 1 Hz; +8 ms: p=0.0034, ON = 13.03 ± 1.03, OFF = 13.7 ± 1.03). (B) Suppressing feedback using ChR2 had no effect on IC spontaneous activity or tone-evoked response magnitude.

The online version of this article includes the following figure supplement(s) for figure 3:

**Figure supplement 1.** Effects of activating cortico-collicular feedback on noise correlations in IC.
**Figure supplement 2.** Effects of modulating cortico-collicular feedback in AC on activity in AC in two example units.

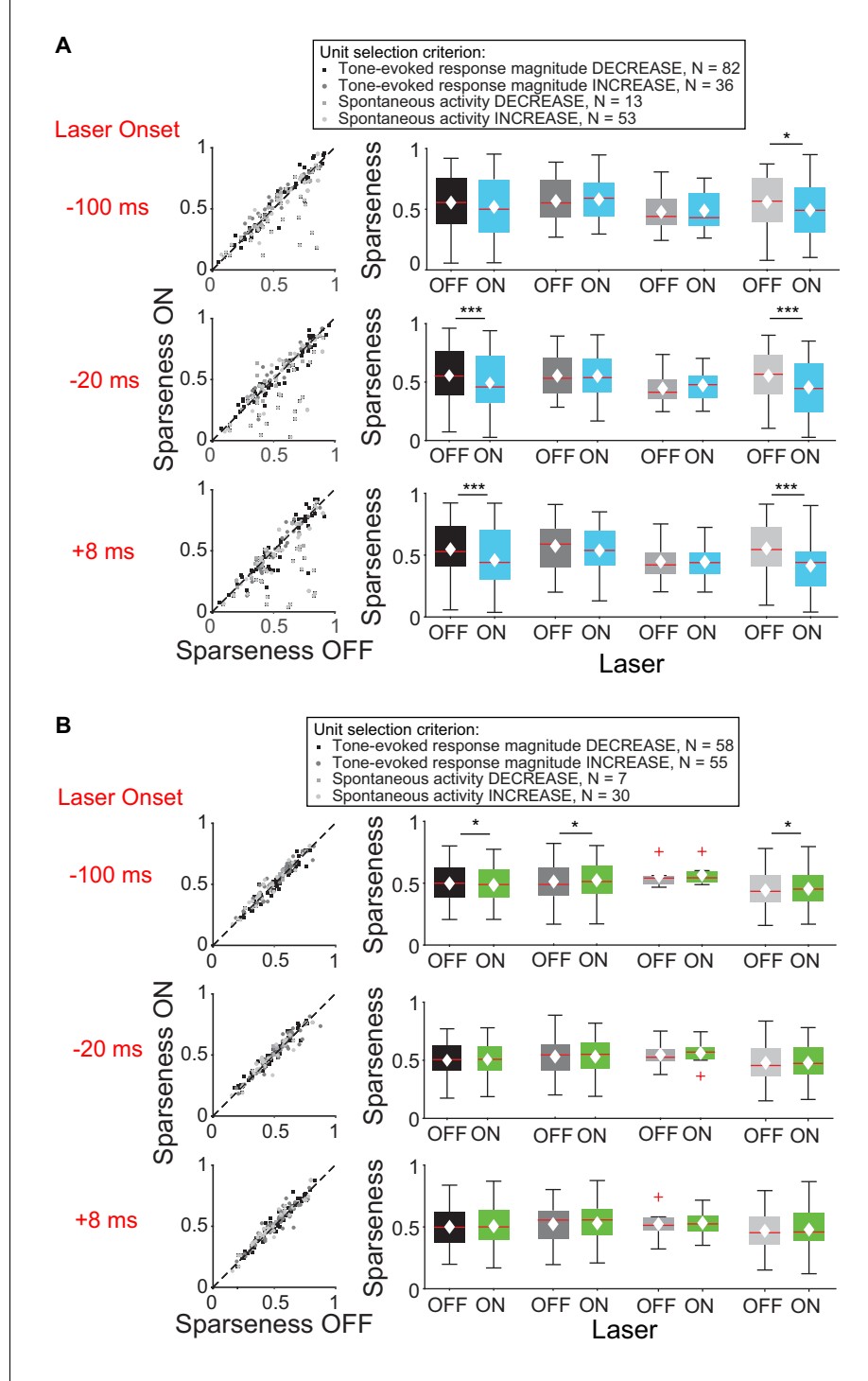

**Figure 4.** Effect of modulating direct cortico-collicular feedback at different latencies relative to tone onset on frequency selectivity in IC. On box plots, red line indicates median, white ◆ indicates mean. (**A**) Left panels: Sparseness for IC units on laser ON (activation with ChR2) versus laser OFF trials. Right panels: Average sparseness for laser ON and laser OFF trials. Activating feedback using ChR2 decreased frequency selectivity in a subset of IC units (mag decrease: −20 ms, p=0.00031, sparse ON = 0.49 ± 0.027, sparse OFF = 0.55 ± 0.025; +8 ms, p=0.00029, sparse ON = 0.46 ± 0.027, sparse OFF = 0.55 ± 0.024; spont increase: −100 ms, p=0.011, sparse ON = 0.49 ± 0.032, sparse OFF = 0.56 ± 0.031; −20 ms, p=0.00018, sparse ON = 0.46 ± 0.034, sparse OFF = 0.56 ± 0.029; +8 ms, p=2.2e-6, sparse ON = 0.41 ± 0.032, sparse OFF = 0.55 ± 0.029). (**B**) Left panels: Sparseness for laser ON (suppression with ArchT) versus laser OFF trials. Right panels: Average sparseness for

*Figure 4 continued on next page*

*Figure 4 continued*

laser ON and laser off trials. Suppressing feedback using ArchT has little effect on frequency selectivity in IC (mag decrease: −100 ms, p=0.011, sparse ON = 0.53 ± 0.021, sparse OFF = 0.51 ± 0.021; mag increase: −100 ms, p=0.019, sparse ON = 0.48 ± 0.021, sparse OFF = 0.49 ± 0.021; spont increase: −100 ms, p=0.043, sparse ON = 0.46 ± 0.029, sparse OFF = 0.45 ± 0.029).

significant effects (DCIC – mag decrease −100 ms: p=3.3e-6, ON = 14.8 Hz, OFF = 16.6 Hz; −20 ms: p=6.3e-7, ON = 12.1 Hz, OFF = 12.6 Hz; spont increase −100 ms: p=0.02 ON=3.5 Hz, OFF = 2.5 Hz; −20 ms: p=2.8e-7, ON = 5.1 Hz, OFF = 4.9 Hz; CNIC – mag decrease +8 ms: p=0.0039, ON = 11.7 Hz, OFF = 12.6 Hz).

To elucidate changes in the shape of the frequency response functions in IC with activation of cortico-cortical feedback beyond the general changes in selectivity we observed with changes in sparseness, we fitted linear regressions to light off versus light on firing rates. This analysis can reveal whether changes in frequency selectivity are linear/subtractive, multiplicative/divisive, or some combination of these transformations. For example, a slope of 1 and a positive y-intercept would represent a linear increase in frequency response (equal increase in response across all frequencies). Over the population of IC neurons with decreased tone-evoked response magnitude and/or increased spontaneous activity, the median slopes and y-intercepts of ranked linear fits to frequency responses are less than 1 and above zero, respectively (*Figure 6A*). These results, in combination with the decreased tone-evoked activity observed (*Figure 6A*; *Figure 6B*; *Figure 6C*, −100 ms: p=0.018, ON = 19.01 ± 1.2 Hz, OFF = 18.5 ± 1.2 Hz; +8 ms: p=0.007, ON = 17.3 ± 1.2 Hz; OFF = 16.9 ± 1.2 Hz), indicate that the decrease in frequency selectivity was due to a decrease in response to tones at preferred frequencies, not non-preferred frequencies. This result suggests that the suppressive effect of the cortico-collicular feedback is preferential for higher firing responses. Suppressing feedback resulted in very small (<0.04%) change in sparseness, therefore it did not affect frequency selectivity in IC (*Figure 4B*).

Correlations in trial-to-trial responsiveness of neurons can affect the information encoded by a neuronal population (*Abbott and Dayan, 1999*; *Averbeck et al., 2006*; *Nirenberg and Latham, 2003*; *Shadlen and Newsome, 1998*; *Zohary et al., 1994*). To test whether activation of cortico-collicular feedback affected correlations in trial-to-trial variability of response strength between units recorded simultaneously we measured noise correlations. We found that activating cortico-collicular feedback using ChR2 had no effect on noise correlations during spontaneous activity or tone response (*Figure 3—figure supplement 1*).

To better understand the effect of modulation of cortical activity on spectro-temporal receptive field properties of IC neurons, we presented a continuous signal comprised of dynamic random chords (DRCs) sampled from a uniform distribution of loudness values per frequency bin. The unbiased nature of this stimulus allowed us to estimate the spectro-temporal receptive field of neurons (STRF), which quantifies the dynamics of sound waveform in time and frequency that lead to a neuronal response. During the DRC stimulus, we turned on the laser every other second for 250 ms to activate cortico-collicular projections with ChR2. We found that a subset of IC units reduced their mean DRC-evoked firing rates (N = 56), whereas another subset of IC units increased their mean DRC-evoked firing rate when we activated cortico-collicular feedback (N = 47). We next separately computed the receptive fields for each neuron when laser was off and when laser was on and compared the STRFs for activation with ChR2 (*Figure 7A*). To quantify those changes, we identified the positive (activation) and negative (suppressive) regions in the STRFs and compared them for laser ON and laser OFF conditions. In the subset of neurons whose firing rate increased with laser STRFs changed: only 42% of positive lobes, and 63% of negative lobes persisted with the laser (*Figure 7B*, left). Of those lobes that persisted, for positive lobes, there was on average a decrease in temporal width (p=0.00098, ON = 0.0303 ± 0.0018 s, OFF = 0.037 ± 0.0023 s), frequency selectivity (p=0.00042, ON = 6.7 ± 1.1 Hz, OFF = 9.6 ± 1.8 Hz) and STRF size (p=0.00036, ON = 46.05 ± 6.9 pixels, OFF = 77.5 ± 12.04 pixels), whereas for negative lobes, we did not detect any changes (*Figure 7C*, left). For IC units whose firing rate was decreased, there was a much smaller change in the lobes with cortico-collicular activation, with 78% and 79% of positive and negative lobes persisting, respectively (*Figure 7B*, right). For both positive lobes and negative lobes, the only difference was an increase in the temporal width when laser activated cortico-collicular feedback (*Figure 7C*, right;

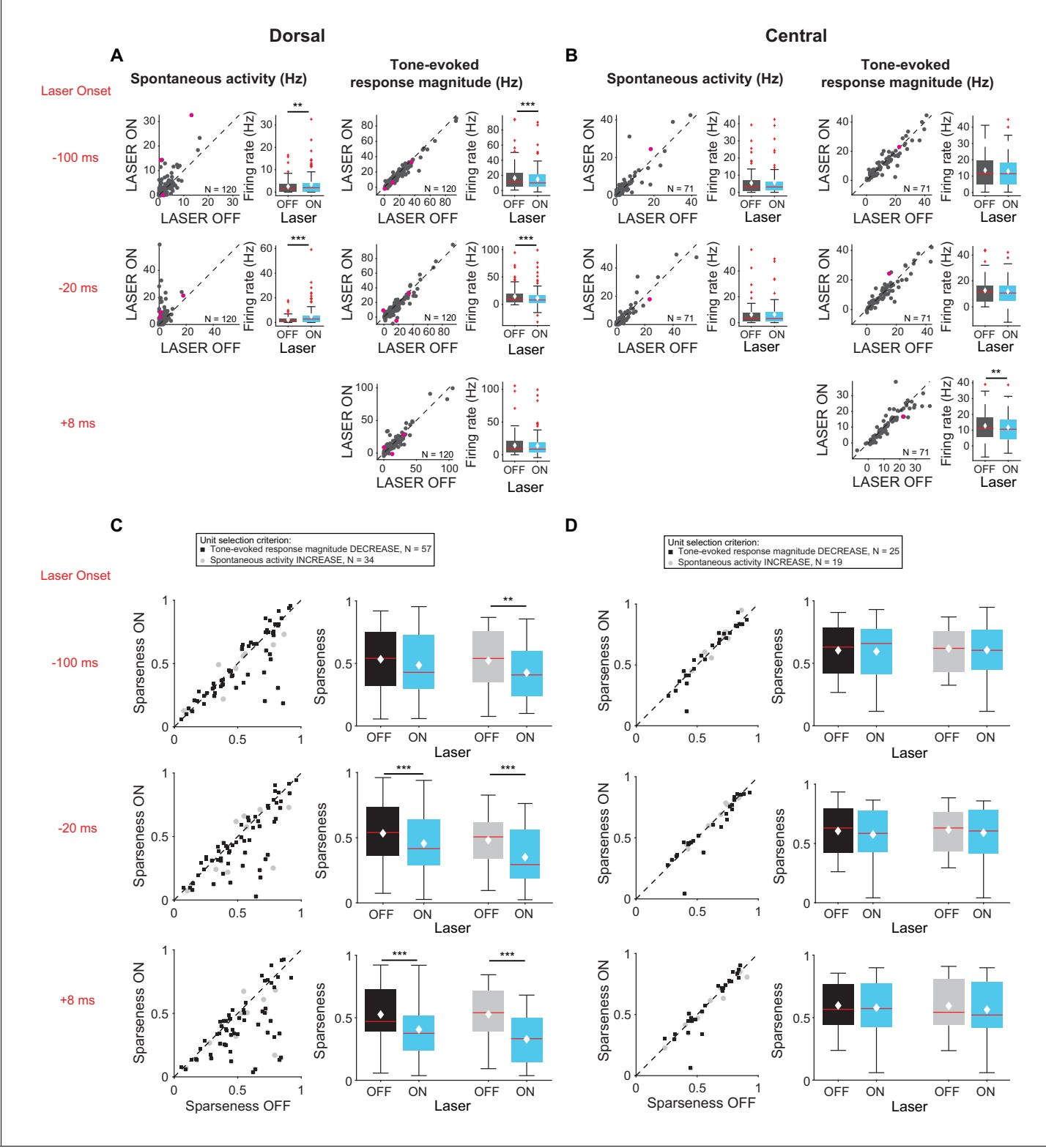

**Figure 5.** Effects of activating cortico-collicular feedback with ChR2 on responses in dorsal and central IC (**A,C**) and DCIC (**B,D**) recording sites. On box plots, red line indicates median, white ◆ indicates mean. (**A**) Increase in spontaneous activity (left; −100 ms: p=0.0204, ON = 3.5 ± 0.5 Hz, OFF = 2.5 ± 0.3 Hz; −20 ms: p=2.8e-7, ON = 5.2 ± 0.7 Hz, OFF = 2.3 ± 0.3 Hz) and decrease in tone-evoked response magnitude (right, −100 ms: p=3.3e-6, ON = 14.8 ± 1.4 Hz, OFF = 16.5 ± 1.5 Hz; −20 ms: p=6.3e-7, ON = 11.8 ± 1.6 Hz, OFF = 14.7 ± 1.4 Hz) for cells in DCIC with activation of cortico-collicular feedback using ChR2. Left panels: neuronal activity (spontaneous, left or tone-evoked, right) on laser ON versus laser off trials. Right *Figure 5 continued on next page*

*Figure 5 continued*

panels: Average neuronal activity (spontaneous, left or tone-evoked, right) for laser on and laser off trials. (B) Little change in spontaneous activity or tone-evoked response magnitude (right; +8 ms: p=0.0027, ON = 13 ± 1.2 Hz, OFF = 13.6 ± 1.2 Hz) for cells in CNIC with activation of cortico-collicular feedback using ChR2. Panels as in A. (C) Activating feedback with ChR2 decreases frequency selectivity in a subset of units (mag decrease: −20 ms, p=6.3e-4, sparse ON = 0.46 ± 0.03, sparse OFF = 0.53 ± 0.03; +8 ms, p=4.7e-5, sparse ON = 0.41 ± 0.03, sparse OFF = 0.53 ± 0.03; spont increase: −100 ms, p=0.0044, sparse ON = 0.43 ± 0.04, sparse OFF = 0.52 ± 0.04; −20 ms, p=0.00013, sparse ON = 0.39 ± 0.04, sparse OFF = 0.53 ± 0.04; +8 ms, p=5.2e-6, sparse ON = 0.33 ± 0.03, OFF = 0.53 ± 0.04) for cells in DCIC. Left panels: Sparseness for laser ON versus laser OFF trials. Right panels: Average sparseness for laser on and laser off trials. (D) Activating feedback has no effect on frequency selectivity of cells in CNIC. Panels as in C.

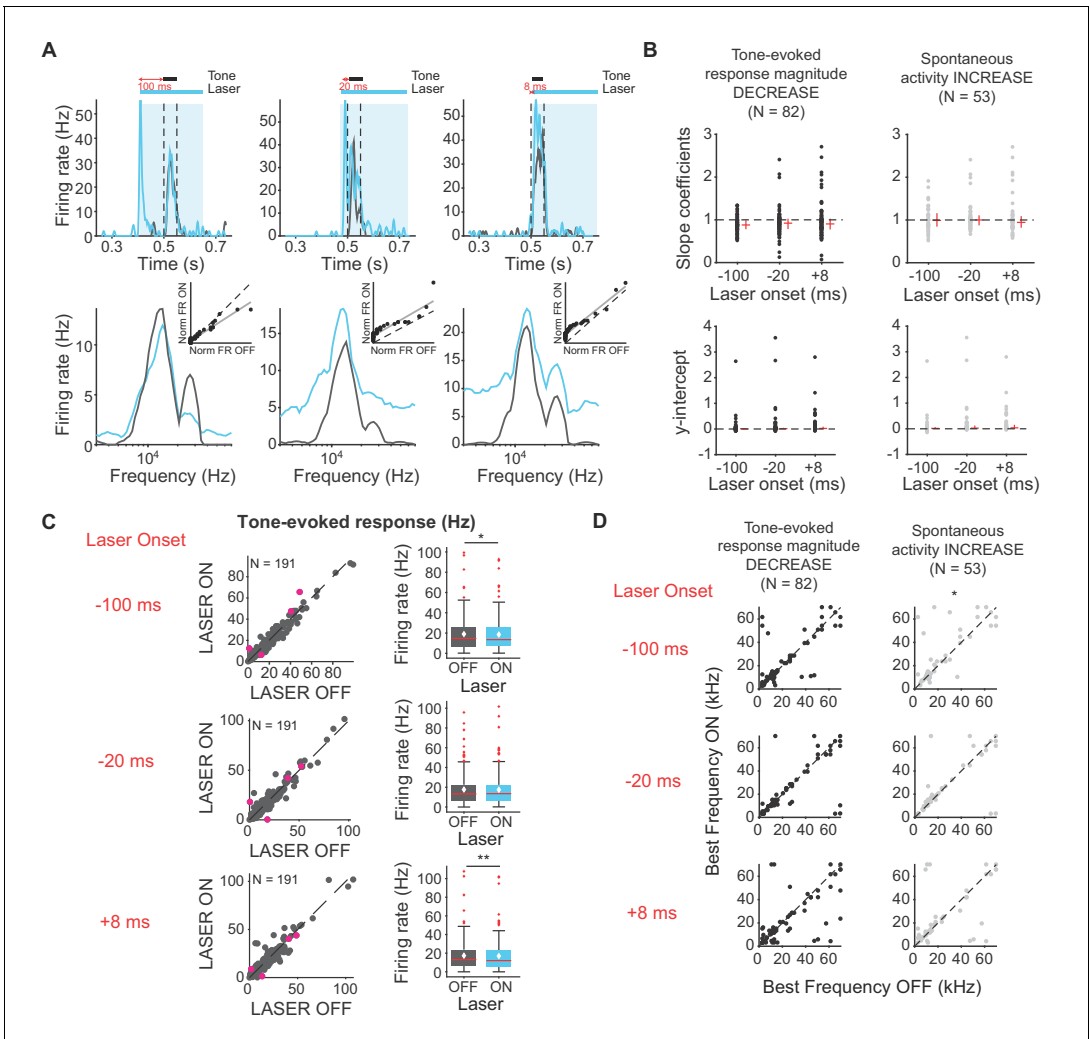

**Figure 6.** Effects of activating direct cortico-collicular feedback on frequency tuning properties of IC neurons. (A) Example IC unit response timecourse (top) and frequency response function (bottom) for laser ON (blue) and laser off (gray) conditions (activation using ChR2). Bottom insets represent ranked linear fits for example unit. (B) Slope coefficients (top, mag decrease: *−100 ms*, median = 0.88; *−20 ms*, median = 0.92; *+8 ms*, median = 0.903; spont increase: *−100 ms*, median = 0.99; *−20 ms*, median = 1; *+8 ms*, median = 0.94) and y-intercepts (bottom, mag decrease: *−100 ms*, median = 0.0086; *−20 ms*, median = 0.014; *+8 ms*, median = 0.022; spont increase: *−100 ms*, median = 0.014; *−20 ms*, median = 0.025; *+8 ms*, median = 0.037) of ranked linear fits of the frequency response function of IC neurons , red horizontal lines indicate median, red vertical lines indicate interquartile range. (C) Left panels: Tone-evoked response for laser ON versus laser off trials. Magenta circles indicate single units. Right panels: Average tone-evoked response for laser ON and laser OFF trials. On box plots, red line indicates median, white ◆ indicates mean. Activating feedback slightly decreased tone-evoked response, averaged across 7 most preferred frequencies (*−100 ms*: p=0.018, ON = 19.01 ± 1.2 Hz, OFF = 18.5 ± 1.2 Hz; *+8 ms*: p=0.007, ON = 17.3 ± 1.2 Hz; OFF = 16.9 ± 1.2 Hz) (D) Best frequencies for laser off versus laser ON trials. Activating feedback had little effect on best frequency (spont increase: *−100 ms*, p=0.022).

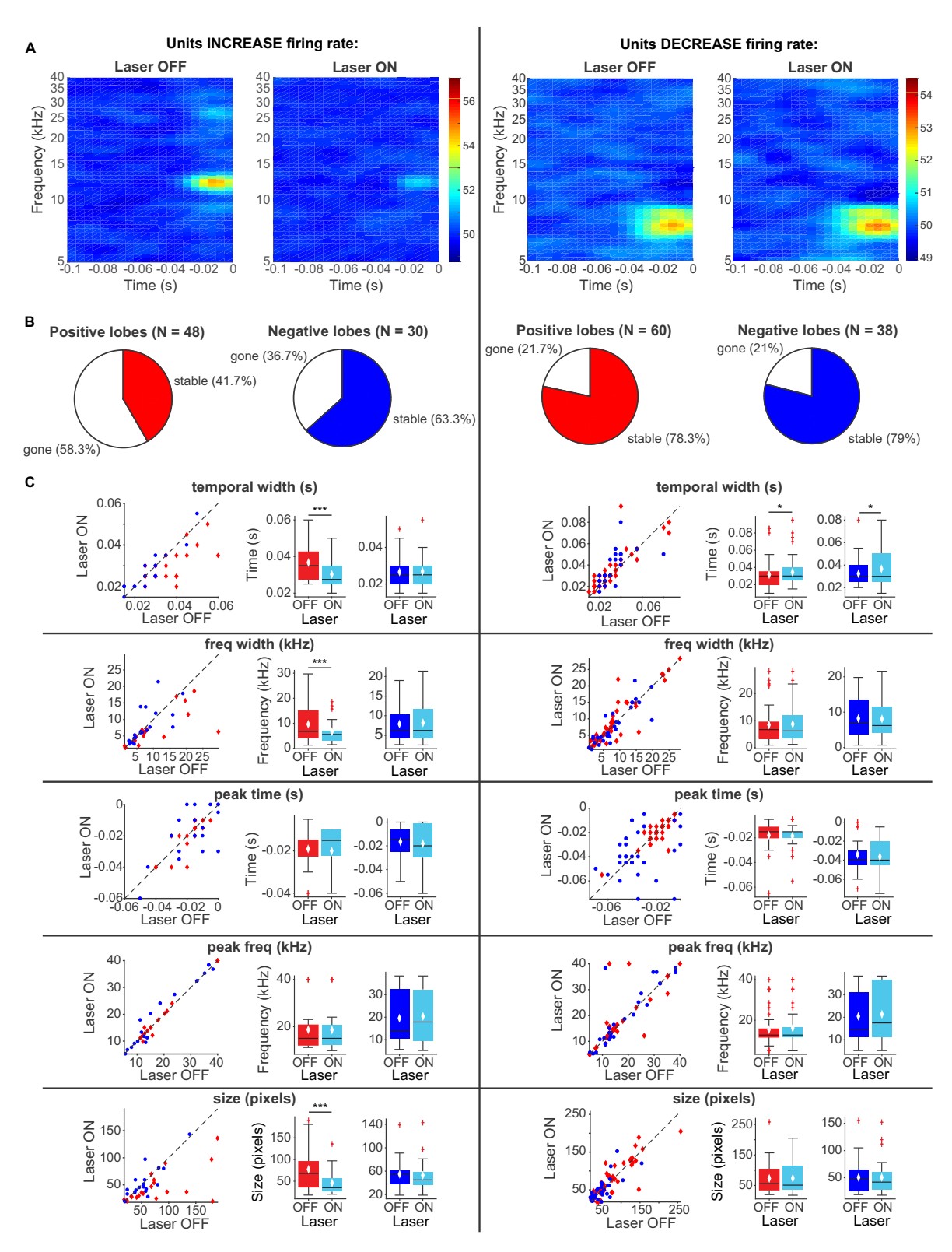

**Figure 7.** Effects of activating direct cortico-collicular feedback using ChR2 on STRF properties of IC units. (**A**) Example STRF with and without activation of feedback for IC units that increase firing rate (left) and decrease firing rate (right) in response to cortico-collicular feedback activation using ChR2. (**B**) Number of positive and negative lobes that persisted with laser (stable). (**C**) Changes in STRF parameters of stable positive and negative lobes. Left panels: STRF parameter for laser OFF versus laser ON trials. Center panels: Average positive lobe STRF parameter for

*Figure 7 continued on next page*

*Figure 7 continued*

laser ON and laser off trials. Right panels: Average negative lobe STRF parameter for laser on and laser off trials. On box plots, red line indicates median, white ◆ indicates mean. For units with an increase in firing rate we observed a decrease in temporal width (p=0.00098, ON = 0.0303 ± 0.0018 s, OFF = 0.037 ± 0.0023 s), frequency width (p=0.00042, ON = 6.7 ± 1.1 Hz, OFF = 9.6 ± 1.8 Hz), and overall size (p=0.00036, ON = 46.05 ± 6.9 pixels, OFF = 77.5 ± 12.04 pixels) of positive lobes. For units with a decrease in firing rate we observed only a small increase in temporal width for both positive (p=0.026, ON = 0.034 ± 0.0025 s, OFF = 0.032 ± 0.0025 s) and negative (p=0.02, ON = 0.037 ± 0.0028 s, OFF = 0.032 ± 0.0024 s) lobes.

Positive lobes: p=0.026, ON = 0.034 ± 0.0025 s, OFF = 0.032 ± 0.0025 s; Negative lobes: p=0.02, ON = 0.037 ± 0.0028 s, OFF = 0.032 ± 0.0024 s). The decrease in STRF size in units with increased DRC-evoked response and decrease in sparseness across the population is consistent with the interpretation that the effect of the cortico-collicular feedback leads to a decrease in responsiveness to tones that evoke the greatest responses (in the center of the receptive field) and an increase to stimuli that evoke weaker activity. In other words, neuronal firing increases overall, but selective responses to specific frequency bands decrease. Because the majority of the units were recorded as multi-units, it is possible that this change in selectivity is due to the change in firing rate of differentially tuned single units.

## Modulating inhibitory neuronal activity in AC does not affect collicular sound responses

Auditory responses in AC are shaped by interactions between excitatory and inhibitory neurons (*Wood et al., 2017*). To determine how modulating frequency selectivity in AC might affect tone-evoked responses in IC in a frequency-selective fashion, we perturbed the excitatory-inhibitory interactions by modulating two different classes of inhibitory interneuron known to contribute to sound responses in AC: PV and SST inhibitory interneurons (*Figure 8A*). We found that modulating PV interneuron activity in AC had no significant effects on spontaneous and tone-evoked activity or frequency selectivity in IC (*Figure 8B–E*) despite modulating frequency selectivity, spontaneous activity, and tone-evoked response magnitude in AC. Specifically, in AC, activating PVs decreased spontaneous activity and tone-evoked response magnitude (*Figure 8F*; spontaneous activity: p=2.2e-9, ON = 0.88 ± 0.16 Hz, OFF = 2.7 ± 0.34 Hz; tone-evoked response magnitude: p=8.02e-7, ON = 6.9 ± 1.06 Hz, OFF = 11.5 ± 1.3 Hz) and increased frequency selectivity (*Figure 8G*; p=3.3e-12, ON = 0.59 ± 0.022, OFF = 0.42 ± 0.02), while suppressing PVs increased spontaneous activity (*Figure 8I*; p=5.2e-5, ON = 4.08 ± 0.54 Hz, OFF = 3.06 ± 0.55 Hz) and decreased frequency selectivity (*Figure 8J*; p=3.8e-5, ON = 0.34 ± 0.027, OFF = 0.41 ± 0.031). This suggests that the changes in cortical activity driven by manipulation of PV activity does not propagate to the inferior colliculus.

Different interneuron classes may function in distinct networks, so we also tested the effects of modulating SST interneurons in AC. Modulating SST interneurons had no effect on tone-evoked activity (*Figure 9B,D*, right) or frequency selectivity (*Figure 9C,E*), but suppressing SST interneurons increased spontaneous activity in IC (*Figure 9D*, left; p=0.029, ON = 6.9 ± 0.66 Hz, OFF = 6.7 ± 0.63 Hz), a change that we also observed with activation of the direct feedback projections (*Figure 2A*, left). Similar to PV interneurons, activating SST interneurons decreased spontaneous activity and tone-evoked response magnitude in AC (*Figure 9F*; spontaneous activity: p=1.4e-12, ON = 0.97 ± 0.25 Hz, OFF = 2.9 ± 0.37 Hz; tone-evoked response magnitude: p=1.3e-15, ON = 3.3 ± 0.59 Hz, OFF = 11.4 ± 1.2 Hz) and increased frequency selectivity (*Figure 9G*; p=1.6e-13, ON = 0.69 ± 0.024, OFF = 0.47 ± 0.019). In AC, suppressing SSTs reduced spontaneous activity, but had no significant effect on tone-evoked response magnitude or frequency selectivity (*Figure 9I, J*; p=8.1e-4, ON = 3.6 ± 0.41 Hz, OFF = 2.2 ± 0.38 Hz). This lack of effect was not due to the relatively small effect of light penetrating to the deep layers. In fact, to confirm that modulating PV and SST activity in AC affected activity of units in L5/6 where the feedback projections we computed changes in spontaneous activity at each tetrode which spanned the entire auditory cortex. We found that activity was modulated across the layers (*Figures 8–9* H,K). Whereas modulating PVs did not have an effect on IC activity, SST suppression resulted in an increase in spontaneous, but not tone-evoked activity in IC, which suggests that inhibitory modulation of sound responses in AC does not propagate to IC.

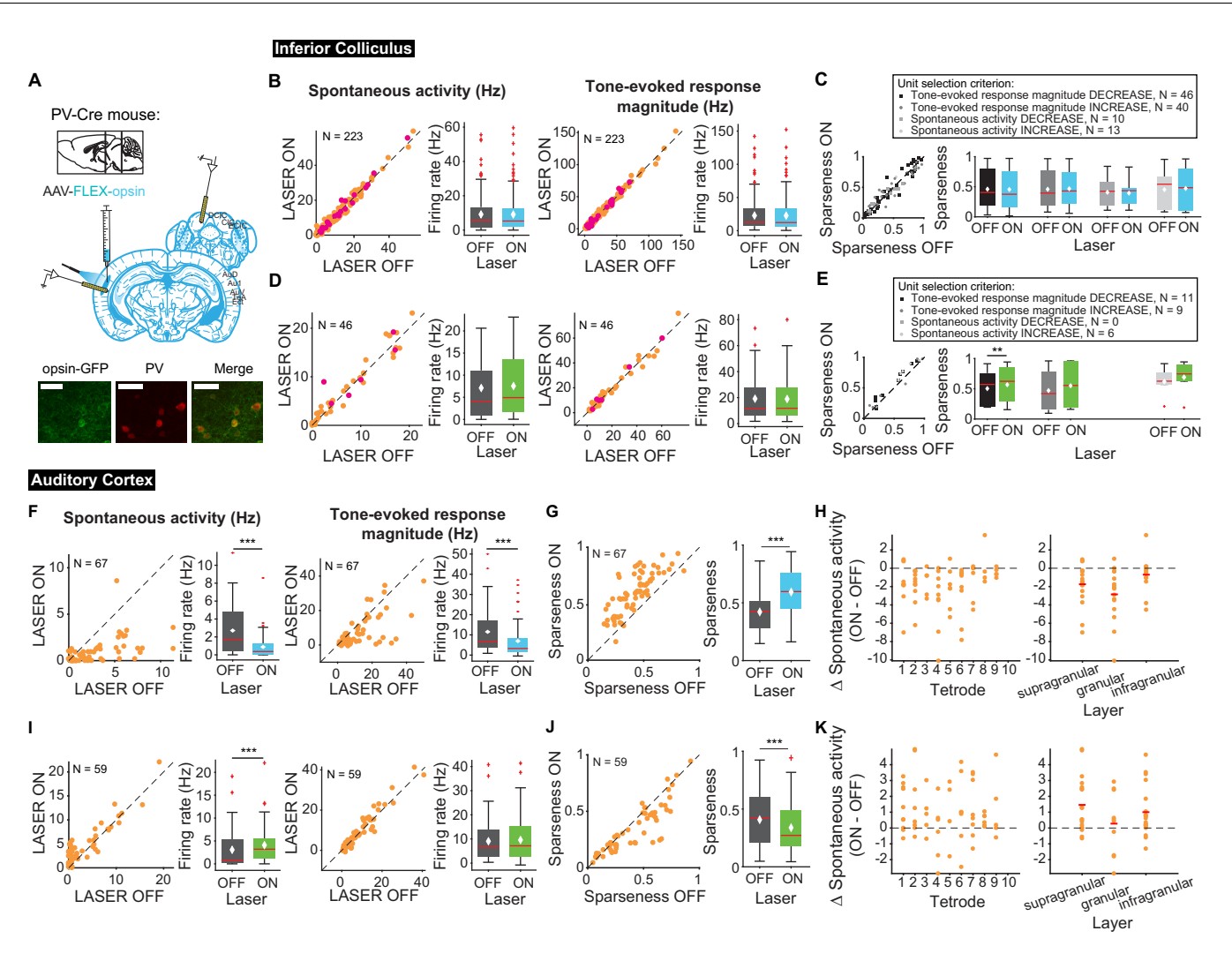

**Figure 8.** Effects of modulating PV interneurons in AC on tone-evoked responses in IC and AC. On box plots, red line indicates median, white ◆ indicates mean. (**A**) Stain for PV (center) and opsin-GFP (left). Scale bar: 50 um. (**B**) Activating PVs had no effect on spontaneous activity or tone-evoked response magnitude in IC. Magenta circles indicate single units. (**C**) Activating PVs had no effect on frequency selectivity in IC. (**D**) Suppressing PVs had no effect on spontaneous activity or tone-evoked response magnitude in IC. Magenta circles indicate single units. (**E**) Suppressing PVs had weak effects on frequency selectivity in IC (mag decrease: p=0.0068, ON = 0.57 ± 0.092, OFF = 0.5 ± 0.087). (**F**) Activating PVs decreased spontaneous activity (p=2.2e-9, ON = 0.88 ± 0.16 Hz, OFF = 2.7 ± 0.34 Hz) and tone-evoked response magnitude (p=8.02e-7, ON = 6.9 ± 1.06 Hz, OFF = 11.5 ± 1.3 Hz) in AC. (**G**) Activating PVs increased frequency selectivity in putative excitatory units in AC (p=3.3e-12, ON = 0.59 ± 0.022, OFF = 0.42 ± 0.02). (**H**) Activating PVs affected putative excitatory units across all layers of AC. (**I**) Suppressing PVs increased spontaneous activity (p=5.2e-5, ON = 4.08 ± 0.54 Hz, OFF = 3.06 ± 0.55 Hz) but did not affect tone-evoked response magnitude in AC. (**J**) Suppressing PVs decreased frequency selectivity in AC (p=3.8e-5, ON = 0.34 ± 0.027, OFF = 0.41 ± 0.031). (**K**) Suppressing PVs affected putative excitatory units across all layers of AC. B,D,F,I Left panels: neuronal activity (spontaneous, left or tone-evoked, right) on laser ON versus laser OFF trials. Right panels: Average neuronal activity (spontaneous, left or tone-evoked, right) for laser ON and laser OFF trials. C,E,G,J Left panels: Sparseness for laser ON versus laser off trials. Right panels: Average sparseness for laser on and laser off trials. (**H,K**) Change in spontaneous activity (laser ON trials – laser off trials), left panels: for units at each tetrode; right panels: separated into supragranular: tetrodes 1–3; granular: tetrodes 5,6; infragranular: tetrodes 7–10.

The online version of this article includes the following figure supplement(s) for figure 8:

**Figure supplement 1.** Effects of activating PV interneurons in AC on spontaneous activity in IC (top) and AC (bottom) in both anesthetized and awake (black edges) preparations.

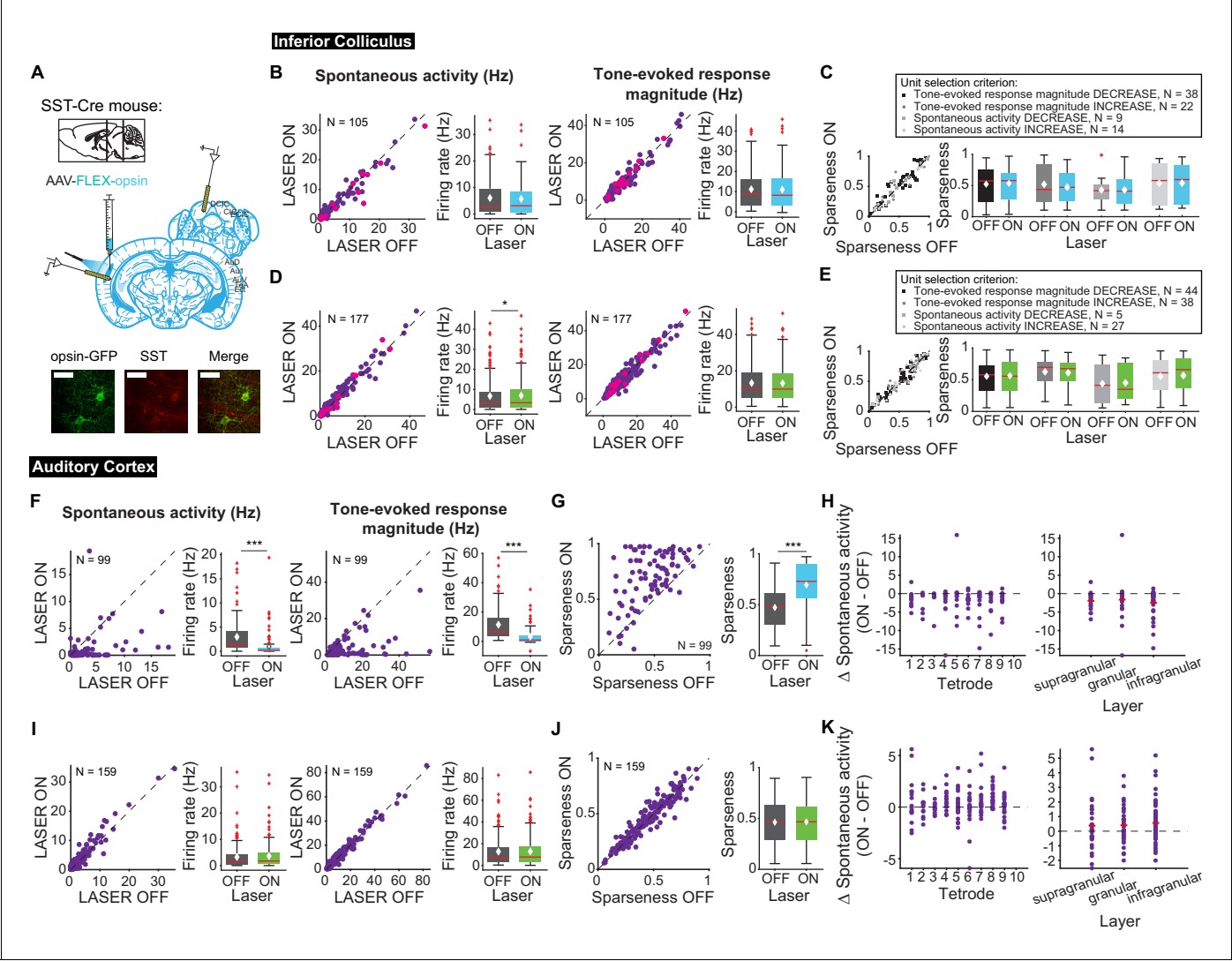

**Figure 9.** Effects of modulating SST interneurons in AC on tone-evoked responses in IC and AC. On box plots, red line indicates median, white ◆ indicates mean. (**A**) Stain for SST (center) and opsin-GFP (left). Scale bar: 50 um. (**B**) Activating SSTs had no effect on spontaneous activity or tone-evoked response magnitude in IC. Magenta circles indicate single units. (**C**) Activating SSTs had no effect on frequency selectivity in IC. (**D**) Suppressing SSTs increased spontaneous activity (p=0.029, ON = 6.9 ± 0.66 Hz, OFF = 6.7 ± 0.63 Hz) but did not affect tone-evoked response magnitude in IC. Magenta circles indicate single units. (**E**) Suppressing SSTs had no effect on frequency selectivity in IC. (**F**) Activating SSTs decreased spontaneous activity (p=1.4e-12, ON = 0.97 ± 0.25 Hz, OFF = 2.9 ± 0.37 Hz) and tone-evoked response magnitude (p=1.3e-15, ON = 3.3 ± 0.59 Hz, OFF = 11.4 ± 1.2 Hz) in AC. (**G**) Activating SSTs increased frequency selectivity in putative excitatory units in AC (p=1.6e-13, ON = 0.69 ± 0.024, OFF = 0.47 ± 0.019). (**H**) Activating SSTs affected putative excitatory units across all layers of AC. (**I**) Suppressing SSTs increased spontaneous activity (p=8.1e-4, ON = 3.6 ± 0.41 Hz, OFF = 2.2 ± 0.38 Hz) but did not affect tone-evoked response magnitude in AC. (**J**) Suppressing SSTs had no effect on frequency selectivity in AC. (**K**) Suppressing SSTs affected putative excitatory units across all layers of AC. B,D,F,I Left panels: neuronal activity (spontaneous, left or tone-evoked, right) on laser on versus laser off trials. Right panels: Average neuronal activity (spontaneous, left or tone-evoked, right) for laser on and laser off trials. C,E,G,J Left panels: Sparseness for laser on versus laser off trials. Right panels: Average sparseness for laser on and laser off trials. (**H,K**) Change in spontaneous activity (laser on trials – laser off trials), left panels: for units at each tetrode; right panels: separated into supragranular: tetrodes 1–3; granular: tetrodes 5,6; infragranular: tetrodes 7–10.

## Discussion

Auditory cortex sends extensive projections to IC (*Bajo and Moore, 2005*; *Bajo et al., 2007*; *Coomes et al., 2005*; *Doucet et al., 2003*; *Saldaña et al., 1996*; *Winer et al., 1998*). Our results demonstrate that activation of this cortico-collicular feedback modulates sound responses in IC by upregulating spontaneous IC activity, decreasing frequency selectivity and reducing spectro-temporal receptive field size in IC (*Figures 2*, *3*, *4*). This effect was localized to DCIC, corresponding to the

sites with largest density of AC projections (*Figure 5*). Interestingly, suppressing cortico-collicular feedback had little effect on IC activity (*Figures 2*, *3*), which suggests that at baseline and during passive tone presentation the feedback does not affect activity of IC neurons. We also found that SST, but not PV inhibitory interneurons modulated IC spontaneous activity, suggesting that the effects of modulation of cortical activity by PVs does not back-propagate to IC, whereas SST-driven modulation affects spontaneous activity, but not tone-evoked responses (*Figures 8*, *9*). Overall our findings imply that direct cortico-collicular feedback can modulate responses to simple (pure tones) and more complex (DRCs) auditory stimuli by reducing, rather than increasing, sound selectivity. This modulation occurs independently of the activity of cortical inhibitory interneurons.

Whereas optogenetic manipulation allows for temporally precise control and cell-type specificity, the technique lacks the spatial specificity of electrical stimulation. Since both AC and cortico-collicular projections are tonotopically organized (*Barnstedt et al., 2015*; *Lim and Anderson, 2007*; *Markovitz et al., 2013*; *Straka et al., 2015*), spatially specific stimulation can more accurately mimic frequency-specific responses by activating specific regions in the tonotopic map. Previous studies found that electrical stimulation of AC caused best frequencies in IC to shift towards the best frequencies of the stimulated site in AC (*Gao and Suga, 1998*; *Gao and Suga, 2000*; *Ma and Suga, 2001a*; *Yan et al., 2005*; *Yan and Suga, 1998*; *Zhou and Jen, 2007*). These results suggest that cortico-collicular feedback may be important for increasing the representation of behaviorally relevant stimuli in IC. However, when activating direct feedback, we did not observe consistent changes in best frequency of IC neurons (*Figure 6D*). This may be due to activation of feedback projections across cortex, and therefore across the tonotopic map, in this paradigm.

Previous studies found no effect of activating projection terminals on responses in the CNIC, only shell regions of IC (*Xiong et al., 2015*), which are the predominant targets of cortico-collicular feedback. In our study we were able to record from a larger neuronal population, revealing a subset of neurons in central IC that were modulated by feedback. This is consistent with another study which observed enhanced white noise-induced activity in a small subset of neurons in the CNIC with activation of direct cortico-collicular feedback (*Vila et al., 2019*). The effects in central IC may be due to direct cortico-collicular feedback, intra-collicular circuits (*Malmierca et al., 1995*; *Miller et al., 2005*; *Saldaña and Merchán, 1992*; *Sturm et al., 2014*), or a combination of these mechanisms.

Previous inactivation studies have shown mixed effects on sound-evoked responses in IC, but consistently demonstrated no effect on frequency selectivity, agreeing with our results. Specifically, suppression of AC increased or decreased IC sound responses in distinct subsets of cells (*Popelár et al., 2003*; *Popelář et al., 2016*) while suppression of direct cortico-collicular feedback terminals in IC decreased sound-evoked responses (*Xiong et al., 2015*). It is possible that we saw little effect from suppression of cortico-collicular projections because inactivation through the auditory cortex suppressed only a subset of feedback projections to IC. Combined, these studies suggest that at baseline the cortex provides only weak modulation of IC activity.

Understanding how AC modulates sound responses in central IC may provide insight into how AC drives changes in auditory behavior. Previously, we found that behavioral frequency discrimination acuity changed after differential auditory fear conditioning and these changes were driven by the auditory cortex (*Aizenberg and Geffen, 2013*; *Aizenberg et al., 2015*). The behavioral task used to test frequency discrimination acuity was a modified pre-pulse inhibition (PPI) of the acoustic startle reflex task. Although AC can modulate this behavior, PPI is still observed after decerebration (*Li and Frost, 2000*) and the underlying circuit is believed to be subcortical (*Fendt et al., 2001*). The IC is a critical structure in the PPI circuit (*Fendt et al., 2001*; *Leitner and Cohen, 1985*; *Li et al., 1998*), with the central nucleus of IC receiving ascending auditory projections and the external nucleus of IC acting as the output station (*Fendt et al., 2001*; *Li and Yue, 2002*). The broad frequency tuning in the shell regions of the IC (*Barnstedt et al., 2015*; *Syka et al., 2000*) made the central nucleus of IC, which has sharper tuning and a tonotopic organization (*Barnstedt et al., 2015*; *Ehret et al., 2003*; *Malmierca et al., 2008*; *Syka et al., 2000*), a good candidate for how AC may drive changes in frequency discrimination acuity. Limited evidence for a descending pathway from AC to the pedunculopontine tegmental nucleus (*Schofield and Motts, 2009*) suggests another alternative pathway by which AC modulates frequency discrimination.

We activated cortico-collicular neurons by shining light in the cortex, thus initiating spikes at the level of the cell body as would occur in response to a sound. An important consideration with this method, however, is that these neurons do not only send collaterals to IC, but also to the auditory

thalamus (*Asokan et al., 2018*; *Williamson and Polley, 2019*), amygdala (*Asokan et al., 2018*), and striatum (*Asokan et al., 2018*). There is also evidence of descending projections to IC from thalamus (*Winer et al., 2002*) and amygdala (*Marsh et al., 2002*). When interpreting our results, it is important to consider that the effects we observe from activating cortico-collicular feedback may also be due, at least in part, to these secondary pathways. One caveat is that we used a square pulse for optogenetic manipulations, whereas a more complex stimulation pattern potentially can achieve stronger activation (*Vila et al., 2019*).

One caveat of our study is that our experiments were performed in passively listening mice. It is possible that the lack of effect we observe with suppression of cortico-collicular feedback is due to the lack of task engagement. Previous studies have found that inactivation of this pathway affects innate responses to sound (*Xiong et al., 2015*) and auditory learning (*Bajo et al., 2010*). Furthermore, studies have found that task engagement modulates neuronal responses to sounds (*Downer et al., 2017*; *Fritz et al., 2003*; *Kuchibhotla et al., 2017*; *Lakatos et al., 2013*; *McGinley et al., 2015*). Thus, we would expect suppression of this pathway to affect sound processing in IC during a behavioral task. Future studies should explore how activity changes with an animal actively engaged in a behavioral task.

We observed little effect of modulating AC inhibitory interneurons on activity in IC despite the changes observed in AC (*Figures 8* and *9*), which suggests there may be another neuron subtype that plays a modulatory role during cortico-collicular plasticity in driving feedback modulation. One possibility that was not explored here are cholinergic inputs from the nucleus basalis (NB). NB, a cholinergic nucleus in the basal forebrain, has been implicated as a key structure in learning-induced auditory plasticity. Studies have found that pairing NB stimulation with presentation of a tone has been sufficient to induce receptive field plasticity in AC (*Kilgard and Merzenich, 1998*; *Kilgard et al., 2001*; *Weinberger, 2003*) and that blocking muscarinic receptors in AC impairs this plasticity (*Miasnikov et al., 2008*). Of greater interest is the receptive field plasticity also observed in IC. Acetylcholine applied to AC increased spontaneous and tone-evoked activity in both AC and IC (*Ji et al., 2001*). Furthermore, pairing AC stimulation with nucleus basalis stimulation (*Ma and Suga, 2003*) or pairing tone presentation with nucleus basalis stimulation (*Zhang et al., 2005*) shifted best frequencies in both AC and IC and application of atropine in AC inhibited this IC plasticity (*Zhang et al., 2005*). It is plausible that NB cholinergic inputs may be responsible for driving responses of IC neurons rather than PV or SST inhibitory interneurons. However, SSTs in AC are depolarized by application of cholinergic agonist (*Kuchibhotla et al., 2017*). While the lack of effect we observe with interneuron inhibition might suggest that SST and PV interneurons are not involved in this circuit, we must also consider that direct activation of feedback is not exactly equivalent to disinhibition. Direct activation causes synchronized activation of cells, while disinhibition allows for asynchronous activation. This difference may explain the different results. Furthermore, the lack of effect we observe with inhibitory interneuron activation is consistent with our results from suppressing cortico-collicular feedback. Thus, while another neuronal subtype may be responsible for driving feedback activity in a behavioral context, inhibitory interneurons may still play a role in shaping responses of feedback once active.

# Materials and methods

## Key resources table

| Reagent type (species) or resource | Designation | Source or reference | Identifiers | Additional information |
|---|---|---|---|---|
| Strain, strain background (*Mus musculus*) | *Pvalb-Cre* mice | Jackson Laboratories | B6;129P2-Pvalbtm1(cre)Arbr/J RRID:IMSR_JAX:008069 | |
| Strain, strain background (*Mus musculus*) | *Sst-Cre* mice | Jackson Laboratories | Ssttm2.1(cre)Zjh/J RRID:IMSR_JAX:013044 | |

*Continued on next page*

*Continued*

| Reagent type (species) or resource | Designation | Source or reference | Identifiers | Additional information |
|---|---|---|---|---|
| Strain, strain background (*Mus musculus*) | Wild type | Jackson Laboratories | C57BL/6J RRID:IMSR_JAX:000664 | |
| Strain, strain background (*Mus musculus*) | *Cdh23* mice | Jackson Laboratories | Cdh23tm2.1Kjn/J RRID:IMSR_JAX:018399 | |
| Recombinant DNA reagent | AAV9-CAG-FLEX-ChR2-tdTomato | Penn VectorCore | Addgene 18917 | |
| Recombinant DNA reagent | AAV9-CAG-FLEX-ArchT-GFP | UNC Vector Core | Addgene 99039 | |
| Recombinant DNA reagent | AAV9-CAG-FLEX-ArchT-tdTomato | UNC Vector Core | Addgene 28305 | |
| Recombinant DNA reagent | RetroAAV2 hSyn Cre-GFP | In house | | Vector generated and maintained in the di Biasi lab |
| Software, algorithm | Matlab | Mathworks | Mathworks.com RRID:SCR_001622 | |
| Software, algorithm | ImageJ | NIH | RRID:SCR_003070 | |

## Animals

All experiments were performed in adult male and female mice (supplier: Jackson Laboratories; age, 12–15 wk; weight, 22–32 g; wild-type C57BL/6J RRID:IMSR_JAX:000664; *Pvalb-Cre* mice, strain: *B6;129P2-Pvalbtm1(cre)Arbr/J* RRID:IMSR_JAX:008069; *Sst-Cre* mice, strain: *Ssttm2.1(cre)Zjh/J* RRID:IMSR_JAX:013044; *Cdh23* mice, strain: *Cdh23tm2.1Kjn/J* RRID:IMSR_JAX:018399, or *Pvalb-Cre* x *Cdh23* or *Sst-Cre* x *Cdh23* crosses). Mice were housed at 28°C on a 12 hr light–dark cycle with water and food provided ad libitum, less than five animals per cage. In *Pvalb-Cre* mice Cre recombinase (Cre) was expressed in parvalbumin-positive interneurons, and in *Sst-Cre*, Cre was expressed in somatostatin-positive interneurons. All animal work was conducted according to the guidelines of University of Pennsylvanian IACUC (protocol number 803266) and the AALAC Guide on Animal Research. Anesthesia by isoflurane and ketamine and euthanasia by CO2 were used. All means were taken to minimize the pain or discomfort of the animals during and following the experiments. All experiments were performed during the animals' dark cycle. Original spike data and code are available on Dryad (http://doi.org/10.5061/dryad.1t61c80).

## Viral vectors

Modified AAVs encoding ArchT (AAV9-CAG-FLEX-ArchT-GFP or AAV9-CAG-FLEX-ArchT-tdTomato; UNC Vector Core) or ChR2 (AAV9-CAG-FLEX-ChR2-tdTomato; Penn Vector Core) were used for selective suppression or excitation, respectively. Retrograde AAV virus encoding Cre (retro AAV-hSyn-Cre-GFP) was custom made in our laboratory. Briefly, RetroAAV2 hSyn Cre-GFP was packaged using the Helper-Free system (Agilent) and the retrograde trafficking plasmid, Retro2, which bears capsid mutations in serotype 2.

## Surgery and virus injection

At least 21 days prior to electrophysiological recordings, mice were anesthetized with isoflurane to a surgical plane. The head was secured in a stereotactic holder. The mouse was subjected to a small craniotomy (2 × 2 mm) over AC under aseptic conditions. Viral particles were injected (750 nl) bilaterally using a syringe pump (Pump 11 Elite, Harvard Apparatus) targeted to AC (coordinates relative to bregma: −2.6 mm anterior,±4.3 mm lateral, +1 mm ventral). Fiber-optic cannulas (Thorlabs, Ø200 μm Core, 0.22 NA) were implanted bilaterally over the injection site at depth of 0.5 mm from the skull surface. For a subset of mice, to target direct feedback, the mouse was also subjected to a craniotomy over IC (1 × 4 mm). Retro AAV viral construct was injected (3 × 200 nl) via glass syringe

(30–50 um diameter) using a syringe pump (Pump 11 Elite, Harvard Apparatus) bilaterally in IC. Craniotomies were covered with a removable silicon plug. A small headpost was secured to the skull with dental cement (C and B Metabond) and acrylic (Lang Dental). For postoperative analgesia, slow release Buprenex (0.1 mg/kg) and Bupivicane (2 mg/kg) were injected subcutaneously. An antibiotic (5 mg/kg Baytril) was injected subcutaneously daily (for 4 days) at the surgical site during recovery. Virus spread was confirmed in all mice postmortem by visualization of the fluorescent protein expression in fixed brain tissue, and its colocalization with PV or SST or, for feedback cohort, expression in AC layer 5/6, following immuno-histochemical processing with the appropriate antibody.

## Confocal imaging

To confirm the location of viral injection sites, tissue sections containing the IC or AC were cut at 40 μm using a cryostat. Sections were mounted onto glass slides and imaged using Zeiss LSM 800 confocal microscope. 20X images were taken throughout the entire IC or AC, and tiles were stitched together to form a composite image using Zen software. Masks were drawn around the edge of the tissue to hide the embedding medium using Photoshop.

## Acoustic stimuli

Stimuli were delivered via a magnetic speaker (Tucker-Davis Technologies) directed toward the mouse's head. Speakers were calibrated prior to the experiments to $\pm$ 3 dB over frequencies between 3 and 70 kHz by placing a microphone (Brüel and Kjaer) in the location of the ear contralateral to the recorded AC hemisphere, recording speaker output and filtering stimuli to compensate for acoustic aberrations (*Carruthers et al., 2013*). Free-standing speaker was used to approximate the conditions under which the mouse typically experiences sounds in awake state.

## Clicks

To obtain a quick assessment of auditory onset responses we used click trains. Click trains were composed of six 50 ms clicks at 70 dB sound pressure level relative to 20 microPascals (SPL) with a 50 ms ISI. Click trains were repeated 120 times with a 450 ms ISI between trains. Alternating click trains were also paired with 1 s laser stimulation beginning 250 ms prior to click train onset.

## Frequency tuning stimuli

To measure tuning for direct feedback cohorts, a train of 50 pure tones of frequencies spaced logarithmically between 3 and 70 kHz, at 70 dB SPL in pseudo-random order was presented 20 times. Each tone was 50 ms duration (5 ms cosine squared ramp up and down) with an inter-stimulus interval (ISI) of 450 ms. Alternating tones were paired with continuous 250 ms laser pulse at either −100 ms, −20 ms, or +8 ms onset relative to tone onset.

For *Sst-Cre* and *Pvalb-Cre* cohorts, a train of pure tones of 35 frequencies spaced logarithmically between 3 and 70 kHz and 8 uniformly spaced intensities from 0 to 70 dB SPL were presented 10 times in a pseudo-random order. Alternating tones were paired with continuous 250 ms laser pulse at −100 ms relative to tone onset.

## Dynamic Random Chords (DRCs)

To measure spectro-temporal receptive fields (STRFs) we constructed DRCs from 20 ms chords (with 1 ms ramp) of 50 frequencies spaced logarithmically between 5 and 40 kHz with average intensity of 50 dB SPL and 20 dB SPL standard deviation selected from a uniform distribution for each chord. Total duration was 40 min with a 250 ms continuous laser pulse presented every 1 s.

## Electrophysiological recordings

All recordings were carried out inside a double-walled acoustic isolation booth (Industrial Acoustics). Mice were placed in the recording chamber, and a headpost was secured to a custom base, immobilizing the head. Activity of neurons in AC were recorded via a custom silicon multi-channel probe (Neuronexus), lowered in the area targeting AC via a stereotactic instrument following a craniotomy at a 35-degree angle. The electrode tips were arranged in a vertical fashion that permits recording the activity of neurons across the depth of the auditory cortex and the inferior colliculus. Activity of neurons in IC were recorded via the same custom probes, lowered in the area targeting IC via a

stereotactic instrument following either a craniotomy (*Sst-Cre* and *Pvalb-Cre* cohorts) or removal of the silicon plug, vertically. Recording sites in IC were identified based on anatomical markers and stereotaxic coordinates as dorsal cortex (<0.5 mm of midline) or central nucleus of IC (center of central nucleus of IC ~1 mm lateral of midline). Recordings were made in multiple locations across IC. Electro-physiological data from 32 channels were filtered between 600 and 6000 Hz (spike responses), digitized at 32 kHz and stored for offline analysis (Neuralynx). Spikes belonging to single neurons and multi-units were detected using commercial software (Plexon). Original spike data and code are available on Dryad (http://doi.org/10.5061/dryad.1t61c80). We examined the experimental groups described in *Table 1* (note: we do not distinguish here between *Sst-Cre/Pvalb* Cre and *Sst-Cre* x *Cdh23* and *Pvalb-Cre* x *Cdh23*). Initial experiments were performed under anesthesia to control for stability or recordings. For cohorts with awake and anesthetized recordings data were analyzed separately, but we observed no difference in our results so data were combined (*Figure 8—figure supplement 1*). We used power analysis for effect size of 25% and expected variance in firing rate (50%) to determine the minimum total number of units for each experimental group at N = 44. All groups included a greater number of units (single- and multi-units were combined in analysis). The number of units in measuring tone-evoked responses are specified in *Table 1*. Mice that did not show effect of laser activation or suppression in auditory cortex were excluded. We recorded from AC in Cdh23+ChR2 mice and Cdh23+ArchT mice to verify an effect in AC (*Figure 3—figure supplement 2*) in all mice used in our analysis.

## Photostimulation of neuronal activity

Neurons were stimulated by application of continuous light pulse delivered from either blue (473 nm, BL473T3-150, used for ChR2 stimulation) or green DPSS laser (532 nm, GL532T3-300, Slocs lasers, used for ArchT stimulation) through implanted cannulas. Laser power measured through cannulas was 3 mW. Timing of the light pulse was controlled with microsecond precision via a custom control shutter system, synchronized to the acoustic stimulus delivery.

## Neural response analysis
### Unit selection

Units were selected based on pure-tone responsiveness. For each unit we identified the 7 frequencies that elicited the highest response and averaged activity across these trials (and the highest 3 amplitudes for stimulus with multiple amplitudes). Units with tone-evoked activity (75 ms window after tone onset) less than two standard deviations above the spontaneous activity (50 ms window prior to tone onset) in no laser condition were excluded from the analysis. Both single units and high quality multi-units (multi-units with <1% of spikes with <1 ms inter-spike-interval) were used.

**Table 1.** Number of recorded units and mice used in our analyses for each experimental group and condition.

| Group | Recording area | Single units | Multi units | Awake total mice | Awake total units | Anesthetized total mice | Anesthetized total units | Total units |
|---|---|---|---|---|---|---|---|---|
| Cdh23-ChR2 | IC | 4 | 187 | 5 Male | 191 | - | - | 191 |
| Cdh23-ArchT | IC | 24 | 181 | 4 Male | 205 | - | - | 205 |
| PV-ChR2 | IC | 25 | 198 | 4 Male | 85 | 12 Male | 138 | 223 |
| | AC | 3 | 64 | | 14 | | 53 | 67 |
| PV-ArchT | IC | 6 | 40 | 4 Male | 46 | - | - | 46 |
| | AC | 3 | 56 | | 59 | | - | 59 |
| SST-ChR2 | IC | 15 | 90 | 7 Male | 76 | 8 Male | 29 | 105 |
| | AC | 9 | 90 | | 49 | | 50 | 99 |
| SST-ArchT | IC | 16 | 161 | 2 Male 2 Female | 140 | 4 Male 2 Female | 37 | 177 |
| | AC | 4 | 155 | | 123 | | 36 | 159 |

## Spontaneous activity and tone-evoked response magnitude

Feedback cohort: Spontaneous activity was the average firing rate in a 20 ms window (to allow for equivalent comparison between −100 ms and −20 ms laser onset conditions) prior to tone onset of top 7 preferred frequencies.

*Sst-Cre/Pvalb-Cre* cohorts: Spontaneous activity was the average firing rate in a 50 ms window prior to tone onset of top 7 preferred frequencies and 3 highest amplitudes.

All mice: Tone-evoked response magnitude was calculated as the difference between the average tone-evoked response in a 75 ms window after tone onset and the spontaneous activity.

*Sparseness.* To examine frequency selectivity of neurons, sparseness of frequency tuning was computed as:

$$Sparseness = 1 - \frac{\left(\sum_{i=1}^{i=n}\frac{FR_i}{n}\right)^2}{\sum_{i=1}^{i=n}\frac{FR_i^2}{n}}$$

where $FR_i$ is tone-evoked firing rate response to tone at frequency $i$, and $n$ is number of frequencies used (*Weliky et al., 2003*). Subgroups of neurons used in sparseness analyses were separated based on changes in spontaneous activity or tone-evoked response magnitude in response to cortico-collicular activation or suppression. Units were selected if > 1 standard deviation change in spontaneous activity or tone-evoked response magnitude based on the -100 ms laser onset trials.

## Linear fits across frequencies

Linear fits were calculated using linear regression (fitlm.m; MATLAB) over 50 data points, one for each of the 50 frequencies tested (*Natan et al., 2017a*). The 50 data points were separately calculated as the mean FR over all repeats of each frequency.

## Best Frequency

Best frequency was defined as the frequency that elicited the maximum response.

## STRF Analysis

To calculate the STRF we separated the stimulus into 1 s chunks, concatenating the 250 ms laser ON chunks and the 250 ms laser OFF chunks immediately preceding laser onset. These data were then used to calculate the average spectrogram preceding a spike. To allow for finer temporal resolution of the STRFs we upsampled the DRCs using nearest neighbor interpolation. Subsequently we averaged the STRF across the eight stimulus files. To determine the significance of the lobes, the *z*-score of pixels was computed relative to the baseline values from an STRF generated with scrambled spike trains, using Stat4ci toolbox (*Chauvin et al., 2005*; *Natan et al., 2017b*). We ran this significance test 100 times and any pixel identified as significant more than 90 times was considered significant. Lobes were matched between laser ON and laser OFF trials by comparing the overlap of the lobes, requiring a 50% overlap of the smallest lobe size to be a match. From the STRF, the peak time, temporal width, peak frequency, and frequency width of the positive and negative lobes were measured (*Schneider and Woolley, 2010*; *Shechter and Depireux, 2007*; *Woolley et al., 2006*).

## Noise correlation

Noise correlations were calculated as the pairwise Pearson correlation coefficient between the spike counts of units recorded simultaneously. For spontaneous activity spikes were counted in a window 20 ms before tone onset and for tone response spikes were counted in a window 75 ms after tone onset.

## Statistical analyses

Significant differences and *P* values were calculated using paired Wilcoxon sign-rank test (unless noted otherwise) with standard MATLAB routine. For the laser alone data, to compare distributions to standard normal distribution data were normalized by mean and standard deviation and then significant differences and *P* values were calculated by Kolmogorov-Smirnoff test with standard

MATLAB routine. Mean ± standard error of the mean was reported unless stated otherwise. * indicates $p < 0.05$, ** indicates $p < 0.01$, *** indicates $p < 0.001$.

## Acknowledgements

This work was supported by National Institutes of Health (Grant numbers NIH R03DC013660, NIH R01DC014779, NIH R01DC015527, NIH R01NS113241), Klingenstein Award in Neuroscience, Human Frontier in Science Foundation Young Investigator Award and the Pennsylvania Lions Club Hearing Research Fellowship to MNG. MNG is the recipient of the Burroughs Wellcome Award at the Scientific Interface. JMB was supported by NIH T32MH017168. We thank the members of the Geffen laboratory and the Hearing Research Center at the University of Pennsylvania, including Dr. Steve Eliades and Dr. Yale Cohen for comments on an earlier version of the manuscript. We also thank Sarah Kwon for assistance with confocal imaging.

## Additional information

### Funding

| Funder | Grant reference number | Author |
|---|---|---|
| National Institute on Deafness and Other Communication Disorders | R03DC013660 | Maria N Geffen |
| National Institute on Deafness and Other Communication Disorders | NIH R01DC014779 | Maria N Geffen |
| National Institute on Deafness and Other Communication Disorders | NIH R01DC015527 | Maria N Geffen |
| Human Frontier Science Program | Young Investigator Award | Maria N Geffen |
| Pennsylvania Lions Club Hearing Research Fellowship | | Maria N Geffen |
| Klingenstein Award in Neurosciences | | Maria N Geffen |
| National Institute of Mental Health | 1F32MH120890 | Alexandria MH Lesicko |
| National Institutes of Health | R01NS113241 | Maria N Geffen |

The funders had no role in study design, data collection and interpretation, or the decision to submit the work for publication.

### Author contributions

Jennifer M Blackwell, Maria N Geffen, Conceptualization, Data curation, Software, Formal analysis, Validation, Investigation, Visualization, Methodology, Writing - original draft, Writing - review and editing; Alexandria MH Lesicko, Formal analysis, Validation, Investigation, Visualization, Methodology; Winnie Rao, Investigation, Visualization, Methodology; Mariella De Biasi, Resources, Visualization

### Author ORCIDs

Jennifer M Blackwell (iD) https://orcid.org/0000-0001-8653-019X
Maria N Geffen (iD) https://orcid.org/0000-0003-3022-2993

### Ethics

Animal experimentation: All animal work was conducted according to the guidelines of University of Pennsylvanian IACUC (protocol number 803266) and the AALAC Guide on Animal Research. Anesthesia by isoflurane and ketamine and euthanasia by $CO_2$ were used. All means were taken to

minimize the pain or discomfort of the animals during and following the experiments. All experiments were performed during the animals' dark cycle.

## Decision letter and Author response
Decision letter https://doi.org/10.7554/eLife.51890.sa1
Author response https://doi.org/10.7554/eLife.51890.sa2

## Additional files

### Supplementary files
• Transparent reporting form

### Data availability
Original spike data and code are available on Dryad (http://doi.org/10.5061/dryad.1t61c80).

The following dataset was generated:

| Author(s) | Year | Dataset title | Dataset URL | Database and Identifier |
|---|---|---|---|---|
| Blackwell JM, Lesicko A, Rao W, De Biasi M, Geffen MN | 2019 | Data from: The role of feedback from the auditory cortex in shaping responses to sounds in the inferior colliculus | http://doi.org/10.5061/dryad.1t61c80 | Dryad Digital Repository, 10.5061/dryad.1t61c80 |

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
