## [Decision Letter]

**Acceptance summary:**

Sensory processing is typically studied along an ascending hierarchy of brain areas, often ignoring feedback between these areas. The auditory cortex sends an extensive feedback to the inferior colliculus, and the authors have found that activation of cortico-collicular feedback reduces sound selectivity in the inferior colliculus by increasing neuronal firing overall, and by attenuating responses in preferred frequency bands. Interestingly, this modulation occurs independently of the activity of cortical inhibitory interneurons. Suppression of this feedback also had no effect on sound-evoked responses in the inferior colliculus, suggesting that during passive sound presentation feedback does not shape activity in the inferior colliculus.

**Decision letter after peer review:**

[Editors’ note: the authors submitted for reconsideration following the decision after peer review. What follows is the decision letter after the first round of review.]

Thank you for submitting your work entitled "The role of feedback from the auditory cortex in shaping responses to sounds in the inferior colliculus" for consideration by *eLife*. Your article has been reviewed by three peer reviewers, including Catherine Emily Carr as the Reviewing Editor and Reviewer #1, and the evaluation has been overseen by a Senior Editor. The following individual involved in review of your submission has agreed to reveal their identity: Sarah Woolley (Reviewer #3).

Our decision has been reached after consultation between the reviewers. Based on these discussions and the individual reviews below, we regret to inform you that your work will not be considered further for publication in *eLife*.

The major concerns of the reviewers were similar, and focused on controls that we think you could carry out with more extensive analyses of your existing data set. We recognize major strengths of the study, which are that your results provide an interesting and detailed picture of how descending input affects IC activity, and align with previous findings.

Our consensus decision is therefore that you should carry out the controls, measurements and analyses needed to address the issues highlighted below. We think this would take too long for a major revision; however, we encourage you to resubmit a manuscript that addresses the points below.

Reviewer #1:

This paper uses optogenetic modulation of cortico-collicular feedback projections to examine sound responses in mouse IC. In addition to activation of this feedback, the paper included experiments to modulate parvalbumin-positive and somatostatin positive inhibitory interneurons. Their main conclusion was that activation of excitatory projections, but not inhibition-driven increases in cortical activity, affected collicular sound responses.

The findings are interesting and surprising. Previous studies had found that electrical stimulation of AC caused best frequencies in IC to shift towards the best frequencies of the stimulated site in AC, and the authors suggest that this may be because the optogenetic stimulation activated whole frequency map, rather than the differently selective electrical stimulation used earlier.

Major concerns: The results are reported as a population that lumps single and multiunit responses, although clear that there are multiple cell types, and IC divisions. The authors observe wide variety of responses to activation of cortico-collicular inputs, shown in the figures. I am less concerned about lumping single and multi-unit responses (although a comparison of the two would be nice), and more concerned about differentiating between the dorsal and central divisions of the IC, since the cortical projection is more prominent in the dorsal regions. The authors state they were able to record from a subset of neurons in central IC, but further support for where they recorded would be useful.

Reviewer #2:

This study examined the effects of several optogenetic manipulations of activity in the auditory cortex (AC) on various measures of neuronal functionality in the inferior colliculus (IC).

One major issue with the study is that the stated goal was to examine the physiological role of projections from the AC to the IC. The first approach taken was then to activate or suppress cortico-collicular projections optogenetically. In order to infer from this what role these projections might play, it is important that these manipulations affect inputs to the IC in a physiologically plausible way. A number of steps would need to be taken in order to determine if this is the case. Firstly, it is not clear how sparse the expression of the retrograde virus is in the AC. Figure 1B suggests that the opsins could be expressed very sparsely, but there is no quantification. Secondly, optogenetic activation provides highly synchronised activation, the magnitude of which may or may not be within physiological ranges. If the manipulations lead to changes in cortico-collicular activity that are not within these ranges, it is not clear what conclusion can be drawn regarding the role of these inputs generally. The second part of the study that approaches the same question by manipulating the activity of inhibitory subpopulations in the cortex would seem to be a better approach. Effectively disinhibiting the cortex should have the effect of asynchronously activating excitatory input to the IC, but in this case little to no effect was observed in the IC.

A second major issue is related to the statistical analyses employed. The first section of the paper claims that activating direct cortico-collicular feedback modulates activity in the IC. The initial experimental approach used was to optogenetically activate or suppress IC-projecting neurons in the AC and measure change in spontaneous activity in the inferior colliculus. It would then make sense to conduct a statistical analysis that addresses the question of whether the spontaneous rate of neuronal units in the IC is different in the laser ON conditions vs the laser OFF conditions. Instead, the authors have chosen to use a Kolmogorov-Smirnoff test to ask when the distribution of change in firing rate was normally distributed. However, it is not possible to conclude from this that the cortical activation actually has caused a significant change. Instead, it would be appropriate to conduct a dependant-samples t-test for laser ON vs laser OFF firing rates (or indeed a Wilcoxon sign-rank test if these data are not normally distributed, as is applied elsewhere in the manuscript). Looking at the data presented in Figure 1E suggests that this would be unlikely to find any significant changes. If anything, the central tendency of the distribution of changes is in the opposite direction to that concluded (and expected) by the authors: in Figure 1E, peaks in the distribution appear to be slightly below zero. The mean and median values appear to be skewed away from this peak by a very small number of units showing high deviations in firing rate.

Reviewer #3:

This study consists of two major groups of experiments investigating the effects of descending inputs from the auditory cortex (AC) to the inferior colliculus (IC) on spontaneous and sound-evoked activity in IC neurons. First, viral vectors are used to specifically infect AC neurons that project to IC with opsins that excite or suppress AC-IC neurons, and the effects of exciting or suppressing these neurons on IC spontaneous and sound-evoked activity are measured. Second, interneuron-class-specific manipulations in AC are used to test whether inhibitory activity in AC impacts IC spontaneous and sound-evoked activity. The goals of the study are largely exploratory/descriptive, rather than hypothesis-driven. With this in mind, the results are informative despite many negative results. Overall, these experiments show that IC activity is influenced by AC-IC input in very specific ways; spontaneous firing in IC units increases with AC-IC input and the linear mapping of IC responses to specific stimulus frequencies is disrupted by AC-IC input. Important methodological and interpretation-related concerns need to be addressed, however.

1) The effects of optogenetic excitation and suppression of AC neurons projecting to IC on IC responses were small to none (Figures 1-3). These results prompt a few methodological concerns.

A) Is it possible that the viral infection methods used to specifically label AC-IC neurons resulted in low rates of infection? This would limit the effects that AC manipulation could have on activity in the IC.

B) Is it possible that the light stimulation did not reach layer 5 with enough power to effectively manipulate AC-IC neurons? This is addressed for the PV and SOM interneuron manipulations, but not for the AC-IC projection neurons.

C) It isn't clear from the Results that the recorded IC neurons were in the spatial location of AC-IC termination.

D) Based on Figures 1, AC stimulation significantly affected a small number of IC units, resulting in small or no average differences between the light on and light off conditions. Were effects of AC manipulation on frequency tuning analyzed separately for these units?

E) The use of response magnitude (difference between evoked and spontaneous rates) to measure effects of AC manipulation on tone evoked responses could explain why tone-evoked responses appeared to decrease with AC stimulation because spontaneous activity increased. Were firing rates evoked by the same frequencies with laser on and laser off compared for each neuron? This would avoid conflation with the laser on-laser off differences in spontaneous activity.

2) The results of the experiments using tones and clicks and those of experiments using the dynamic random chord (DRC) stimulus are difficult to reconcile. Figure 4 shows what appear to be large effects of AC stimulation on IC receptive fields (STRFs). Given this, the explanation for the disappearance of significant positive and negative lobes with laser stimulation should be pursued. For example, if the interpretation given at the end of subsection “Activating direct cortico-collicular feedback modulates activity in the inferior colliculus” should be tested by going back to the tone-evoked responses and measuring effects of AC stimulation on responses to tones outside of the 7 frequencies that evoked the largest responses from a unit.

3) So much work clearly went into the PV and SOM interneuron manipulations and recordings in Ac and IC. The results of those experiments and interpretation of the results should be further explained. For example, why were the effects of interneuron manipulation observed in AC but not IC units?

[Editors’ note: further revisions were suggested prior to acceptance, as described below.]

Thank you for resubmitting your work entitled "The role of feedback from the auditory cortex in shaping responses to sounds in the inferior colliculus." for further consideration by *eLife*. Your revised article has been evaluated by Barbara Shinn-Cunningham (Senior Editor) and a Reviewing Editor.

The manuscript has been improved but there are some remaining issues that need to be addressed before acceptance, as outlined below:

Summary

The manuscript had been previously reviewed at *eLife*, and rejected because the reviewers felt that the controls, measurements and analyses needed to address the issues raised by reviewers would take too long for a major revision; however, we encouraged resubmission. The new manuscript has been reviewed, and we felt that the study and topic area merit publication in *eLife*.

Nevertheless, several important caveats to the approach and analyses remain, which could be addressed in the Materials and methods and/or Discussion. We will attach the relevant reviews in addition to the summary here, and suggest that you acknowledge and address the following points in your revised submission. Other thoughtful suggestions follow, and we include the reviews with suggestions. The other reviewers had no major changes.

Major changes:

1) That there are potential limitations of using rectangular laser pulses as described previously (Vila et al., 2019)

2) That in spite of the cell-specific expression there hypothetically could be mixed suppressive and excitatory effects that are lost in multi-unit spike rate and spectro-temporal receptive field (STRF) analyses used here.

3) There are also concerns about normality in spike rate distributions. With respect to the spike rate effects in the new Figure 5, panel A (dorsal IC, -100 ms laser condition), if the spike rate distributions are not normal then the bar graphs could be replaced with box-plots emphasizing that there are indeed spike-rate outliers. Note that the appropriate statistical comparisons should be performed.

Here are the full reviews that include suggestions for revision. The main points are summarized above.

Reviewer #1:

1) In their revision, the authors have sidestepped one of my major concerns, namely: "Secondly, optogenetic activation provides highly synchronised activation, the magnitude of which may or may not be within physiological ranges. If the manipulations lead to changes in cortico-collicular activity that are not within these ranges, it is not clear what conclusion can be drawn regarding the role of these inputs generally." At the very least, the authors should explicitly acknowledge the limitations of using rectangular laser pulses in this context and the associated limitations to the generality of their conclusions. For example, a recent study addressing the same questions has shown that these effects in the IC strongly depend on patterns of laser activation in the AC (Vila et al., 2019).

2) A new important issue has appeared in the revised manuscript, which merits careful consideration. The new Figure 5 is used to support the conclusion that "…units recorded from dorsal recording sites are significantly affected by activation of feedback, but centrally located units are not". However, the firing rates of some dorsal units are twice as high as the central ones'. Compare the scales in panels A and B. Could the effect in, for example, -100 ms tone-evoked "dorsal" be due to those high-firing-rate units being suppressed by the laser? One can clearly see that those data points are below the diagonal. It would be crucial to reanalyse the data presented in panel A, where the firing rates are up to 100 Hz, by using only those units whose firing rates are comparable to those in panel B, i.e., less than 40 Hz. Otherwise, the conclusion might reflect the effect of the laser on the small number of relatively rare units with high firing rates, disproportionately suppressed by the laser. As those cells are relatively rare (about 5%?), they might have been missed in the central IC (71 units; versus 120 units in the dorsal IC). Without this analysis, I am not convinced by the authors' conclusion.

3) The authors have indicated in the revision letter which data points corresponded to single units (red dots). This information is very useful. Yet they chose not to indicate it in the revised manuscript. Why? I think it must be indicated.

4) In general, my concern with multi-unit data is that it is possible, given the bidirectional effects of the laser stimulation, that some nearby neurons are modulated positively and other neurons are modulated negatively. If these neurons are recorded as a multiunit, then the total effect will be a weighted sum of the individual effects. This will bias not only the determination of the effects on the firing rates, but may also induce artefacts in the calculation of STRFs, unless all individual neurons contributing to the multiunit have the same selectivity. However, since they were not recoded as single units, their individual selectivity is unknown. You can see, for example, that the receptive field in Figure 7A 'laser on" differs significantly from that in the "laser off" condition. These are units that increase their firing rate with the laser on. Which means there are more spikes; yet the STRF almost vanishes. This illustrates what I have said above: what the authors interpret as a change in selectivity could be due to a change in the firing rates of differentially tuned single units, all contributing to the vector sum during STRF calculation, without any change in individual neurons' selectivity.

5) The authors persist in using bars and standard errors for data that are so obviously non-normally distributed.

Reviewer #2:

The authors have thoroughly responded to reviewers' critiques with new analyses, revised figures and additional information in the Materials and methods, Results and Discussion. I am satisfied that this paper will contribute importantly to our understanding of how the auditory system's descending projections shape sound coding.

Reviewer #3:

Clearly cortical feedback to collicular structures including superior and inferior colliculi plays a pivotal role in shaping neural pathway responses and behavior and yet little is known about auditory circuits mediating this process. Prior studies have dumped pharmacological agents on cortex or electrically stimulated cortex to explore the sound processing of these feedback circuits. However, these manipulations are not cell-type specific. This is a rigorous and well-executed study demonstrating for the first time cell-types involved in cortical feedback shaping of sound evoked responses in central and dorsal IC. I have only very minor comments and suggestions to improve the write up as outlined below.

Title, "Role of feedback from auditory cortex in shaping responses to sounds in inferior colliculus"

Title.

1) Shorten title?, e.g. "Shaping inferior colliculus responses to sound through cortical feedback"

Introduction section.

1) Please open with a paragraph explaining why we care about cortico-collicular feedback? You end paragraph two with some discussion of why we may care. Maybe round this out and move up to the top paragraph of Introduction.

2) Introduction paragraph three, add a captivating introduction sentence to the paragraph, e.g., "Cortical output modulations of collicular activity is not a monolithic process but instead is shaped by interactions between excitatory and inhibitory cortical neurons."

3) Introduction paragraph two, you have a long sentence "Whereas inactivation studies, on the other hand, found less consistent effects on IC responses.…) Please simplify. Perhaps, "Prior studies find pharmacological inactivation of AC alters tone-evoked responses of IC neurons (Zhang, 1997) whereas others do not (Jen, 1998)." Is the Jen, 1997 study using pharma to inactivate AC?

Terminology throughout manuscript

1) Non-specialist audiences will not get what the difference is between the dorsal cortex of IC and central nucleus. In parts of the manuscript you refer to central nucleus simply as "IC" whereas others it is "central IC". Please lay this out clearly and refer to all your recordings that are verified as being in Central IC as being in CIC being consistent with your abbreviation for dorsal cortex as DCIC.

Materials and methods.

1) In Materials and methods, clearly explain your rationale for doing these initial experiments under anesthesia e.g., " To obtain precise responses across large numbers of IC neurons, these experiments were carried out under anesthesia. Though cortical spike rate responses are reduced under anesthesia, the general frequency response organization remains the same. "

2) “Acoustic stimuli” subsection, please give your rationale for delivering sound from a free field speaker. Perhaps, one rationale is to more closely approximate conditions animals may experience in awake conditions? Note the caveats. If you are calibrating next to the contralateral ear in free field and then using a filter to compensate for sound reflections -presumably you have a means of insuring that the set-up (stereotax unit, electrode holders, amplifiers etc are in fixed spatial positions relative to the ear and you have distinct filters for left and right ear configurations). Using ear-tubes may be a better approach as the mice have relatively uniform ear canals and calibrating can be more precise.

3) “Acoustic stimuli” section, provide a rationale for including click train responses in your sound battery. Click train responses have been used in the past for ABR and penetrating electrode studies but these sounds are fraught with spectral splatter that makes interpretation of the response relatively difficult. Perhaps, your rationale was to get a quick assessment of auditory onset responses?

“Neural Response Analysis” subsection, you define FRi as "tone-evoked response" here and above (describe that you're assessing firing rate). Can you please explain that Fri is a "tone-evoked firing rate response"? Also please explain the rationale for computing sparseness and the rationale for your 20 ms time scale for doing so. For example, there are prior studies that have compared cortical and collicular time-scales for sparse coding and find that central IC has an order of magnitude shorter integration times (~10ms) than cortex and that on those time scales encoding in IC is indeed sparse (Chen, Read, Escabi, J Neurosci 2012; https://www.ncbi.nlm.nih.gov/pubmed/22723685).

Results section.

1) Results section, first paragraph. The authors are commended for carrying out these important, rigorous and difficult studies with cell specific modulation and e-phys but they should recognize that most of your audience is less informed about theses dual viral techniques. Please do a better job of setting up this paragraph so as not to lose the general audience. Following the first introduction sentence please help orient the non-expert with something like, "A combination of local and retrograde viral transfections were made in order to achieve cell and circuit specific viral transfections of cortico-collicular neurons".

2) In general, most paragraphs in the Results section do a great job of introducing the rationale for experiments and analyses. Thank you!

3) Discussion paragraph three, here you refer to central IC as "central IC" but many other places throughout the manuscript you are ambiguous e.g. Discussion paragraph one, you say things like "…little effect on IC activity" is this central and dorsal IC? Also you say "effects of modulation of cortical activity by PVs does not backpropagate to IC". Do you DCIC here?

---

## [Author Response]

[Editors’ note: the authors resubmitted a revised version of the paper for consideration. What follows is the authors’ response to the first round of review.]

Reviewer #1:[…] Major concerns: The results are reported as a population that lumps single and multiunit responses, although clear that there are multiple cell types, and IC divisions. The authors observe wide variety of responses to activation of cortico-collicular inputs, shown in the figures. I am less concerned about lumping single and multi-unit responses (although a comparison of the two would be nice), and more concerned about differentiating between the dorsal and central divisions of the IC, since the cortical projection is more prominent in the dorsal regions. The authors state they were able to record from a subset of neurons in central IC, but further support for where they recorded would be useful.

This is an important concern, and a very good suggestion. During the experiment, we targeted two recordings in the central nucleus of IC and two recordings in the dorsal cortex of IC (closer to midline). Previously, we combined these recordings together in the analysis. In the revision, we separated the units based on whether the recording site was more central or more dorsal. Indeed, we find that units recorded from dorsal recording sites are significantly affected by activation of feedback, but centrally located units are not.

We have included these results in Figure 5 and the Results: “Interestingly, these changes are observed almost exclusively in cells located in DCIC rather than CNIC (Figure 5). Specifically, activation of feedback increased spontaneous activity (Figure 5A, left; -100 ms: p = 0.0204, ON = 3.5 ± 0.5 Hz, OFF = 2.5 ± 0.3 Hz; -20 ms: p = 2.8e-7, ON = 5.2 ± 0.7 Hz, OFF = 2.3 ± 0.3 Hz) and decreased tone-evoked response magnitude (Figure 5A, right, -100 ms: p = 3.3e-6, ON = 14.8 ± 1.4 Hz, OFF = 16.5 ± 1.5 Hz; -20 ms: p = 6.3e-7, ON = 11.8 ± 1.6 Hz, OFF = 14.7 ± 1.4 Hz) in cells located in DCIC, but CNIC (Figure 5B). Similarly, activation of feedback decreased frequency selectivity (Figure 5C; mag decrease: -20 ms, p = 6.3e-4, sparse ON = 0.46 ± 0.03, sparse OFF = 0.53 ± 0.03; +8 ms, p = 4.7e-5, sparse ON = 0.41 ± 0.03, sparse OFF = 0.53 ± 0.03; spont increase: -100 ms, p = 0.0044, sparse ON = 0.43 ± 0.04, sparse OFF = 0.52 ± 0.04; -20 ms, p = 0.00013, sparse ON = 0.39 ± 0.04, sparse OFF = 0.53 ± 0.04; +8 ms, p = 5.2e-6, sparse ON = 0.33 ± 0.03, OFF = 0.53 ± 0.04) in cells located in DCIC, but not CNIC (Figure 5D). This is consistent with the enhanced density of AC projections in dorsal as compared to central IC (Figure 1D).”

We also amended the Materials and methods to clarify identification of our recording sites “Recording sites in IC were identified based on anatomical markers and stereotaxic coordinates as dorsal cortex (<0.5 mm of midline) or central nucleus of IC (center of central nucleus of IC ~1 mm lateral of midline). Recordings were made in multiple locations across IC.”

Reviewer #2:[…] One major issue with the study is that the stated goal was to examine the physiological role of projections from the AC to the IC. The first approach taken was then to activate or suppress cortico-collicular projections optogenetically. In order to infer from this what role these projections might play, it is important that these manipulations affect inputs to the IC in a physiologically plausible way. A number of steps would need to be taken in order to determine if this is the case. Firstly, it is not clear how sparse the expression of the retrograde virus is in the AC. Figure 1B suggests that the opsins could be expressed very sparsely, but there is no quantification.

We compared the strength of opsin expression in slices in multiple mice, and found that the transfection was as strong as was evident in previously published studies. We quantified the expression by counting the number of cell bodies that were expressing the fluorescent protein. We found that in layer 5 of the auditory cortex, we had ~5 cell bodies/100um x 100um ROI. This level of expression is similar to that observed by Williamson and Polley, 2019, Figure 1A, left. We now include a new figure (Figure 1) that demonstrates the spread of virus expression in IC, the transfected cells in AC, and the labelled projections within IC. Furthermore, we quantified the distribution of projections in IC, which shows that AC projects not just to the external shell, but also to the central nucleus.

Secondly, optogenetic activation provides highly synchronised activation, the magnitude of which may or may not be within physiological ranges. If the manipulations lead to changes in cortico-collicular activity that are not within these ranges, it is not clear what conclusion can be drawn regarding the role of these inputs generally. The second part of the study that approaches the same question by manipulating the activity of inhibitory subpopulations in the cortex would seem to be a better approach. Effectively disinhibiting the cortex should have the effect of asynchronously activating excitatory input to the IC, but in this case little to no effect was observed in the IC.

This is a great point, and we added in the Discussion: “While the lack of effect we observe with interneuron inhibition might suggest that SOM and PV interneurons are not involved in this circuit, we must also consider that direct activation of feedback is not exactly equivalent to disinhibition. Direct activation causes synchronized activation of cells, while disinhibition allows for asynchronous activation. This difference may explain the different results.”

A second major issue is related to the statistical analyses employed. The first section of the paper claims that activating direct cortico-collicular feedback modulates activity in the IC. The initial experimental approach used was to optogenetically activate or suppress IC-projecting neurons in the AC and measure change in spontaneous activity in the inferior colliculus. It would then make sense to conduct a statistical analysis that addresses the question of whether the spontaneous rate of neuronal units in the IC is different in the laser ON conditions vs the laser OFF conditions. Instead, the authors have chosen to use a Kolmogorov-Smirnoff test to ask when the distribution of change in firing rate was normally distributed. However, it is not possible to conclude from this that the cortical activation actually has caused a significant change. Instead, it would be appropriate to conduct a dependant-samples t-test for laser ON vs laser OFF firing rates (or indeed a Wilcoxon sign-rank test if these data are not normally distributed, as is applied elsewhere in the manuscript). Looking at the data presented in Figure 1E suggests that this would be unlikely to find any significant changes. If anything, the central tendency of the distribution of changes is in the opposite direction to that concluded (and expected) by the authors: in Figure 1E, peaks in the distribution appear to be slightly below zero. The mean and median values appear to be skewed away from this peak by a very small number of units showing high deviations in firing rate.

To address this concern, we performed a paired Wilcoxon sign-rank test on the laser on vs laser off firing rate of each unit. We still observe significant increases in firing rate with ChR2 activation of AC-IC feedback. We have updated the Results and Figure 2 legend to reflect the new statistics as well as to include mean and s.e.m. of the changes in firing rate for each significant duration. We also updated Figure 2 to include scatter plots to show the changes in firing rate across the population in addition to the distributions.

Figure 2 C: Changes in spontaneous activity with activation of AC-IC projections (*1 ms*: p = 5.3e-4, ON = 5.6 ± 0.6 Hz, OFF = 4.9 ± 0.5 Hz; *5 ms*: p = 0.0027, ON = 6.9 ± 0.7 Hz, OFF = 4.8 ± 0.5 Hz; *250 ms*: p = 0.005, ON = 5.6 ± 0.6 Hz, OFF = 4.8 ± 0.5 Hz)

Figure 2 D: Changes in spontaneous activity with suppression of AC-IC projections (2*5 ms*: p = 0.039, ON = 9.5 ± 0.7 Hz, OFF = 9.5 ± 0.7 Hz; *250 ms*: p = 0.02, ON = 9.5 ± 0.8 Hz, OFF = 9.5 ± 0.8 Hz)

Reviewer #3:[…] Overall, these experiments show that IC activity is influenced by AC-IC input in very specific ways; spontaneous firing in IC units increases with AC-IC input and the linear mapping of IC responses to specific stimulus frequencies is disrupted by AC-IC input. Important methodological and interpretation-related concerns need to be addressed, however.1) The effects of optogenetic excitation and suppression of AC neurons projecting to IC on IC responses were small to none (Figures 1-3). These results prompt a few methodological concerns.A) Is it possible that the viral infection methods used to specifically label AC-IC neurons resulted in low rates of infection? This would limit the effects that AC manipulation could have on activity in the IC.

Please see response to reviewer #2, question 1.

B) Is it possible that the light stimulation did not reach layer 5 with enough power to effectively manipulate AC-IC neurons? This is addressed for the PV and SOM interneuron manipulations, but not for the AC-IC projection neurons.

We observe effects of light stimulation across all layers in cortex (Figure 5 H,K; Figure 6 H,K). We use identical procedures for cannula implant and laser power for the AC-IC projection cohorts. Amended Materials and methods to … “We recorded from AC in our Cdh23+ChR2 mice and Cdh23+ArchT mice to verify an effect in AC (Figure 3—figure supplement 2) in all mice used in our analysis” … for clarification

C) It isn't clear from the Results that the recorded IC neurons were in the spatial location of AC-IC termination.

Please see response to reviewer #1, question 1.

D) Based on Figures 1, AC stimulation significantly affected a small number of IC units, resulting in small or no average differences between the light on and light off conditions. Were effects of AC manipulation on frequency tuning analyzed separately for these units?

Yes. In Figures 3, 5, and 6, we separately analyze frequency selectivity for different subsets of neurons based on effect of light on condition. We state this in the Results section, for example “Activation of feedback decreased frequency selectivity in the subsets of units that also showed a decrease in tone-evoked response magnitude or increase in spontaneous activity, but not in units that showed an increase in tone-evoked response magnitude or decrease in spontaneous activity”, as well as in the Materials and methods section “Subgroups of neurons used in sparseness analyses were separated based on changes in spontaneous activity or tone-evoked response magnitude in response to cortico-collicular activation or suppression. Units were selected if > 1 standard deviation change based on the -100 ms laser onset trials.”

E) The use of response magnitude (difference between evoked and spontaneous rates) to measure effects of AC manipulation on tone evoked responses could explain why tone-evoked responses appeared to decrease with AC stimulation because spontaneous activity increased. Were firing rates evoked by the same frequencies with laser on and laser off compared for each neuron? This would avoid conflation with the laser on-laser off differences in spontaneous activity.

We compared spontaneous activity, tone-evoked activity, and tone-evoked response magnitude in response to the same frequency for each neuron (chosen as the best frequency in the laser OFF condition), each under laser-ON and laser-OFF conditions. We find the same effects as when averaging across top 7 frequencies in each condition. We find an increase in spontaneous activity (left; *-20 ms*: p = 0.0094, ON = 6.5 ± 0.5, OFF = 4.1 ± 0.9) and decrease in tone-evoked response magnitude (right; *-100 ms*: p = 8.2e-5, ON = 14.7 ± 1.1 Hz; OFF = 17.9 ± 1.3; *-20 ms*: p = 1.6e-6, ON = 12.9 ± 1.4, OFF = 17.6 ± 1.3; *+8 ms*: p = 0.013, ON = 15.3 ± 1.3, OFF = 16.6 ± 1.3), as in Figure 2A in manuscript, as well as a decrease in tone-evoked activity (middle; *-100 ms*: p = 2.8e-4, ON = 19.2 ± 1.3 Hz; OFF = 21.4 ± 1.4; *-20 ms*: p = 3.7e-4, ON = 19.4 ± 1.4, OFF = 21.7 ± 1.4; *+8 ms*: p = 9.8e-5, ON = 19.05 ± 1.4, OFF = 20.9 ± 1.4), as in Figure 3E in manuscript.

2) The results of the experiments using tones and clicks and those of experiments using the dynamic random chord (DRC) stimulus are difficult to reconcile. Figure 4 shows what appear to be large effects of AC stimulation on IC receptive fields (STRFs). Given this, the explanation for the disappearance of significant positive and negative lobes with laser stimulation should be pursued. For example, if the interpretation given at the end of subsection “Activating direct cortico-collicular feedback modulates activity in the inferior colliculus” should be tested by going back to the tone-evoked responses and measuring effects of AC stimulation on responses to tones outside of the 7 frequencies that evoked the largest responses from a unit.

We address changes in activity outside of most preferred frequencies in Figure 6, which examines changes in the shape of the frequency response curve. We find that units decrease response at more preferred frequencies, suggested by the linear fit slopes being less than 1 and supported by Figure 6B, and increase at less preferred frequencies, suggested by linear fit y-intercepts being greater than 1 (further evidence provided Author response image 1).

**Author response image 1. respfig1:** Response averaged across 7 less preferred frequencies (29-35th most preferred frequencies). Activation of direct cortico-collicular feedback increases tone-evoked activity (middle; +8 ms: p = 6.4e-4, ON = 7.1 ± 0.7, OFF = 5.7 ± 0.6) at less preferred frequencies.

For clarification of our interpretation of these results we added “Activation of feedback decreased frequency selectivity in the subsets of units that also showed a decrease in tone-evoked response magnitude or increase in spontaneous activity, but not in units that showed an increase in tone-evoked response magnitude or decrease in spontaneous activity”, as well as in the Materials and methods section “Subgroups of neurons used in sparseness analyses were separated based on changes in spontaneous activity or tone-evoked response magnitude in response to cortico-collicular activation or suppression. Units were selected if > 1 standard deviation change based on the -100 ms laser onset trials.”

3) So much work clearly went into the PV and SOM interneuron manipulations and recordings in Ac and IC. The results of those experiments and interpretation of the results should be further explained. For example, why were the effects of interneuron manipulation observed in AC but not IC units?

To address our interpretation as to why we observe effects in AC, but not IC, with manipulation of interneurons in AC we added to the Discussion … “We observed little effect of modulating AC inhibitory interneurons on activity in IC despite the changes observed in AC (Figures 8, 9), which suggests there may be another neuron subtype that plays a modulatory role during cortico-collicular plasticity in driving feedback modulation. […] Thus, while another neuronal subtype may be responsible for driving feedback activity in a behavioral context, inhibitory interneurons may still play a role in shaping responses of feedback once active.”

[Editors’ note: what follows is the authors’ response to the second round of review.]

Nevertheless, several important caveats to the approach and analyses remain, which could be addressed in the Materials and methods and/or Discussion. We will attach the relevant reviews in addition to the summary here, and suggest that you acknowledge and address the following points in your revised submission. Other thoughtful suggestions follow, and we include the reviews with suggestions. The other reviewers had no major changes.Major changes:1) That there are potential limitations of using rectangular laser pulses as described previously (Vila et al., 2019)

Added to the Discussion: “One caveat to consider in interpreting these results is that we used a square pulse for optogenetic manipulations, whereas a more complex stimulation pattern potentially can achieve stronger activation (Vila et al., 2019).”

2) That in spite of the cell-specific expression there hypothetically could be mixed suppressive and excitatory effects that are lost in multi-unit spike rate and spectro-temporal receptive field (STRF) analyses used here.

Added to the Results: “Because the majority of the units were recorded as multi-units, this change in selectivity may be due to the change in firing rate of differentially tuned single units.”

3) There are also concerns about normality in spike rate distributions. With respect to the spike rate effects in the new Figure 5, panel A (dorsal IC, -100 ms laser condition), if the spike rate distributions are not normal then the bar graphs could be replaced with box-plots emphasizing that there are indeed spike-rate outliers. Note that the appropriate statistical comparisons should be performed.

All bar plots changed to box plots to elaborate the distribution of the data. We used non-parametric statistical methods as described in “Statistical Analyses” where appropriate.

Here are the full reviews that include suggestions for revision. The main points are summarized above.Reviewer #1:1) In their revision, the authors have sidestepped one of my major concerns, namely: "Secondly, optogenetic activation provides highly synchronised activation, the magnitude of which may or may not be within physiological ranges. If the manipulations lead to changes in cortico-collicular activity that are not within these ranges, it is not clear what conclusion can be drawn regarding the role of these inputs generally." At the very least, the authors should explicitly acknowledge the limitations of using rectangular laser pulses in this context and the associated limitations to the generality of their conclusions. For example, a recent study addressing the same questions has shown that these effects in the IC strongly depend on patterns of laser activation in the AC (Vila et al., 2019).

Added to the Discussion: “One caveat to consider in interpreting these results is that we used a square pulse for optogenetic manipulations, whereas a more complex stimulation pattern potentially can achieve stronger activation (Vila et al., 2019).”

2) A new important issue has appeared in the revised manuscript, which merits careful consideration. The new Figure 5 is used to support the conclusion that "…units recorded from dorsal recording sites are significantly affected by activation of feedback, but centrally located units are not". However, the firing rates of some dorsal units are twice as high as the central ones'. Compare the scales in panels A and B. Could the effect in, for example, -100 ms tone-evoked "dorsal" be due to those high-firing-rate units being suppressed by the laser? One can clearly see that those data points are below the diagonal. It would be crucial to reanalyse the data presented in panel A, where the firing rates are up to 100 Hz, by using only those units whose firing rates are comparable to those in panel B, i.e., less than 40 Hz. Otherwise, the conclusion might reflect the effect of the laser on the small number of relatively rare units with high firing rates, disproportionately suppressed by the laser. As those cells are relatively rare (about 5%?), they might have been missed in the central IC (71 units; versus 120 units in the dorsal IC). Without this analysis, I am not convinced by the authors' conclusion.

We added the following to the manuscript: “To verify these results were not due to a select few high firing units found in DCIC, we repeated the analysis comparing only units with spontaneous activity in the laser OFF condition less than 40 Hz. With these matched firing rates we still see the same statistically significant effects (DCIC – mag decrease -100 ms: p = 3.3e-6, ON = 14.8 Hz, OFF = 16.6 Hz; -20 ms: p = 6.3e-7, ON = 12.1 Hz, OFF = 12.6 Hz; spont increase -100 ms: p = 0.02 ON = 3.5 Hz, OFF = 2.5 Hz; -20 ms: p = 2.8e-7, ON = 5.1 Hz, OFF = 4.9 Hz; CNIC – mag decrease +8 ms: p = 0.0039, ON = 11.7 Hz, OFF = 12.6 Hz).”

3) The authors have indicated in the revision letter which data points corresponded to single units (red dots). This information is very useful. Yet they chose not to indicate it in the revised manuscript. Why? I think it must be indicated.

We have added reference to the IC single units in our figures as in the revision letter.

4) In general, my concern with multi-unit data is that it is possible, given the bidirectional effects of the laser stimulation, that some nearby neurons are modulated positively and other neurons are modulated negatively. If these neurons are recorded as a multiunit, then the total effect will be a weighted sum of the individual effects. This will bias not only the determination of the effects on the firing rates, but may also induce artefacts in the calculation of STRFs, unless all individual neurons contributing to the multiunit have the same selectivity. However, since they were not recoded as single units, their individual selectivity is unknown. You can see, for example, that the receptive field in Figure 7A 'laser on" differs significantly from that in the "laser off" condition. These are units that increase their firing rate with the laser on. Which means there are more spikes; yet the STRF almost vanishes. This illustrates what I have said above: what the authors interpret as a change in selectivity could be due to a change in the firing rates of differentially tuned single units, all contributing to the vector sum during STRF calculation, without any change in individual neurons' selectivity.

Added to the Results: “Because the majority of the units were recorded as multi-units, this change in selectivity may be due to the change in firing rate of differentially tuned single units.”

5) The authors persist in using bars and standard errors for data that are so obviously non-normally distributed.

All bar plots changed to box plots to elaborate the distribution of the data.

Reviewer #3:Clearly cortical feedback to collicular structures including superior and inferior colliculi plays a pivotal role in shaping neural pathway responses and behavior and yet little is known about auditory circuits mediating this process. […] I have only very minor comments and suggestions to improve the write up as outlined below.Title, "Role of feedback from auditory cortex in shaping responses to sounds in inferior colliculus"1) Shorten title?, e.g. "Shaping inferior colliculus responses to sound through cortical feedback"

Good point. We shortened the title to “Auditory cortex shapes sound responses in the inferior colliculus”.

Introduction section.1) Please open with a paragraph explaining why we care about cortico-collicular feedback? You end paragraph two with some discussion of why we may care. Maybe round this out and move up to the top paragraph of Introduction.

Amended end of first paragraph to: “Previous studies have demonstrated the importance of cortical feedback to IC in auditory behaviors (Bajo, Nodal, Moore, and King, 2010; Xiong et al., 2015), however, the mechanisms by which information processing is shaped via the *descending feedback* pathway remain poorly characterized.”

2) Introduction paragraph three, add a captivating introduction sentence to the paragraph, e.g., "Cortical output modulations of collicular activity is not a monolithic process but instead is shaped by interactions between excitatory and inhibitory cortical neurons."

Changed first sentence to: “In AC, modulation of sound responses is not a monolithic process, but instead is shaped by interactions between excitatory and inhibitory cortical neurons.”

3) Introduction paragraph two, you have a long sentence "Whereas inactivation studies, on the other hand, found less consistent effects on IC responses…) Please simplify. Perhaps, "Prior studies find pharmacological inactivation of AC alters tone-evoked responses of IC neurons (Zhang, 1997) whereas others do not (Jen, 1998)." Is the Jen, 1997 study using pharma to inactivate AC?

Changed to: “Whereas one study found inactivation of AC caused a shift in best frequency in IC neurons (Zhang, Suga, and Yan, 1997), several other studies showed that inactivation of AC had no effect on frequency selectivity in IC (Jen et al., 1998), but rather modulated sound-evoked and spontaneous activity (Gao and Suga, 1998; J Popelář, Nwabueze-Ogbo, and Syka, 2003; Jiří Popelář et al., 2016)”

We still want to include a bit more information about the previous studies than would be in the suggested simplified sentence, but we reworded for clarity.

Terminology throughout manuscript1) Non-specialist audiences will not get what the difference is between the dorsal cortex of IC and central nucleus. In parts of the manuscript you refer to central nucleus simply as "IC" whereas others it is "central IC". Please lay this out clearly and refer to all your recordings that are verified as being in Central IC as being in CIC being consistent with your abbreviation for dorsal cortex as DCIC.

This has been corrected. Now all uses to IC refer to cases when we are not distinguishing between the different regions.

Materials and methods.1) In Materials and methods, clearly explain your rationale for doing these initial experiments under anesthesia e.g., " To obtain precise responses across large numbers of IC neurons, these experiments were carried out under anesthesia. Though cortical spike rate responses are reduced under anesthesia, the general frequency response organization remains the same."

To Materials and methods – Electrophsiological Recordings we added: “Initial experiments were performed under anesthesia to control for stability or recordings.”

2) “Acoustic stimuli” subsection, please give your rationale for delivering sound from a free field speaker. Perhaps, one rationale is to more closely approximate conditions animals may experience in awake conditions? Note the caveats. If you are calibrating next to the contralateral ear in free field and then using a filter to compensate for sound reflections -presumably you have a means of insuring that the set-up (stereotax unit, electrode holders, amplifiers etc are in fixed spatial positions relative to the ear and you have distinct filters for left and right ear configurations). Using ear-tubes may be a better approach as the mice have relatively uniform ear canals and calibrating can be more precise.

We have added to the stimulus section: “Free-standing speaker was used to approximate the conditions under which the mouse typically experiences sounds in awake state.”

We are very careful at calibrating the speaker. All the parts of the setup are at exact fixed positions relative to the speaker. We use the white noise approach for the calibration procedure. We record the signal at 3 different slightly offset positions of a high quality (B and K) calibration microphone.

3) “Acoustic stimuli” section, provide a rationale for including click train responses in your sound battery. Click train responses have been used in the past for ABR and penetrating electrode studies but these sounds are fraught with spectral splatter that makes interpretation of the response relatively difficult. Perhaps, your rationale was to get a quick assessment of auditory onset responses?

Added: “To obtain a quick assessment of auditory onset responses we used click trains.”

“Neural Response Analysis” subsection, you define FRi as "tone-evoked response" here and above (describe that you're assessing firing rate). Can you please explain that Fri is a "tone-evoked firing rate response"?

Amended to read “tone-evoked firing rate response”

Also please explain the rationale for computing sparseness and the rationale for your 20 ms time scale for doing so. For example, there are prior studies that have compared cortical and collicular time-scales for sparse coding and find that central IC has an order of magnitude shorter integration times (~10ms) than cortex and that on those time scales encoding in IC is indeed sparse (Chen, Read, Escabi, J Neurosci 2012; https://www.ncbi.nlm.nih.gov/pubmed/22723685).

We are using a measure sparseness to measure frequency selectivity within an individual neuron as it is less sensitive to the shape of the frequency response function than gaussian widths (Aizenberg et al., 2015). This is distinct from the “sparseness” definition in the Chen et al. study.

Results section.1) Results section, first paragraph. The authors are commended for carrying out these important, rigorous and difficult studies with cell specific modulation and e-phys but they should recognize that most of your audience is less informed about theses dual viral techniques. Please do a better job of setting up this paragraph so as not to lose the general audience. Following the first introduction sentence please help orient the non-expert with something like, "A combination of local and retrograde viral transfections were made in order to achieve cell and circuit specific viral transfections of cortico-collicular neurons".

Added the following sentence after the introductory sentence of this paragraph to help clarify technique: “A combination of cortical anterograde and collicular retrograde viral transfections in order to achieve specific viral transfection of cortico-collicular neurons.”

2) In general, most paragraphs in the Results section do a great job of introducing the rationale for experiments and analyses. Thank you!

Thank you for your kind comment.

3) Discussion paragraph three, here you refer to central IC as "central IC" but many other places throughout the manuscript you are ambiguous e.g. Discussion paragraph one, you say things like "…little effect on IC activity" is this central and dorsal IC? Also you say "effects of modulation of cortical activity by PVs does not backpropagate to IC". Do you DCIC here?

When we use the abbreviation ‘IC’ we are not distinguishing between CNIC and DCIC. We only use the terms CNIC and DCIC when we are explicitly referring to the recordings where we distinguish between these two areas.